# Scaling Short-Term Memory of Visuomotor Policies for Long-Horizon Tasks

## Abstract

Many robotic tasks demand short-term memory, whether it's retrieving objects that are no longer visible or turning off an appliance after a certain amount of time. Yet, most visuomotor policies remain myopic, relying only on immediate sensory input without leveraging past experiences to guide decisions. We present PRISM, a transformer-based architecture for visuomotor policies to effectively use short-term memory via two key components: (i) gated attention, which selectively filters retrieved information to suppress irrelevant details, and (ii) a hierarchical architecture that first compresses local interactions into compact tokens and then integrates them to capture temporally extended dependencies. Together, these mechanisms enable us to scale short-term memory in visuomotor policies for up to two minutes at five frames per second, an order of magnitude longer than previous approaches. To systematically evaluate memory in visuomotor control, we introduce ReMem-Bench—a benchmark of eight diverse household manipulation tasks spanning four categories of short-term memory—designed to foster general memory mechanisms rather than siloed, task-specific solutions. PRISM consistently outperforms prior works, including transformer-based visuomotor policies with short-term memory, recurrent architectures, and other short-term memory-management strategies. In ReMemBench and real-world evaluation, PRISM achieves an absolute improvement of 11–15 points over the strongest baseline. On RoboCasa and LIBERO, it further yields 11–14-point gains over its no-memory variant and outperforms strong fine-tuned VLA baselines such as GR00T-N1-3B and Open-VLA, without any pretraining. Together, PRISM and ReMemBench establish a foundation for developing and evaluating short-term memory–augmented visuomotor policies that scale to long-horizon tasks. Additional materials are available at `https://remembench-prism.github.io`

## 1 Introduction

Memory is central to intelligent behavior: human cognition depends on retaining past experiences to guide ongoing actions (Squire, 2004; Anderson et al., 2015). Classic studies in human cognition distinguish between sensory, short-term, and long-term memory (Atkinson & Shiffrin, 1968). Among these, short-term memory, which spans seconds to minutes, supports everyday activities such as remembering to stir at regular intervals, retrieving objects that are no longer visible, or counting scoops of salt (Baddeley, 2020). Without it, behavior collapses to responses grounded only in the present observation, unable to account for past experiences.

Recent advances in robot learning have led to the development of generalist visuomotor policies through imitation learning (Black et al., 2024; Bjorck et al., 2025; Shukor et al., 2025). Yet, these policies operate without any short-term memory, relying only on immediate sensory inputs. A key factor is the reliance on a vanilla transformer backbone (Vaswani et al., 2017). Although transformers provide a form of short-term memory by storing information in their context window, their attention mechanism suffers from two drawbacks. First, it is susceptible to irrelevant information in the context window (Hong et al., 2025). This can induce spurious correlations, *e.g.*, linking past distractor locations to a left or right placement decision, degrading test performance (De Haan et al., 2019; Wen et al., 2020). Second, attention computation scales quadratically with context length (Vaswani et al., 2017), making it inefficient to process high-dimensional sensory inputs, such as image obser-

vations, over long temporal sequences. These drawbacks limit the scalability of short-term memory over long horizons in visuomotor policies.

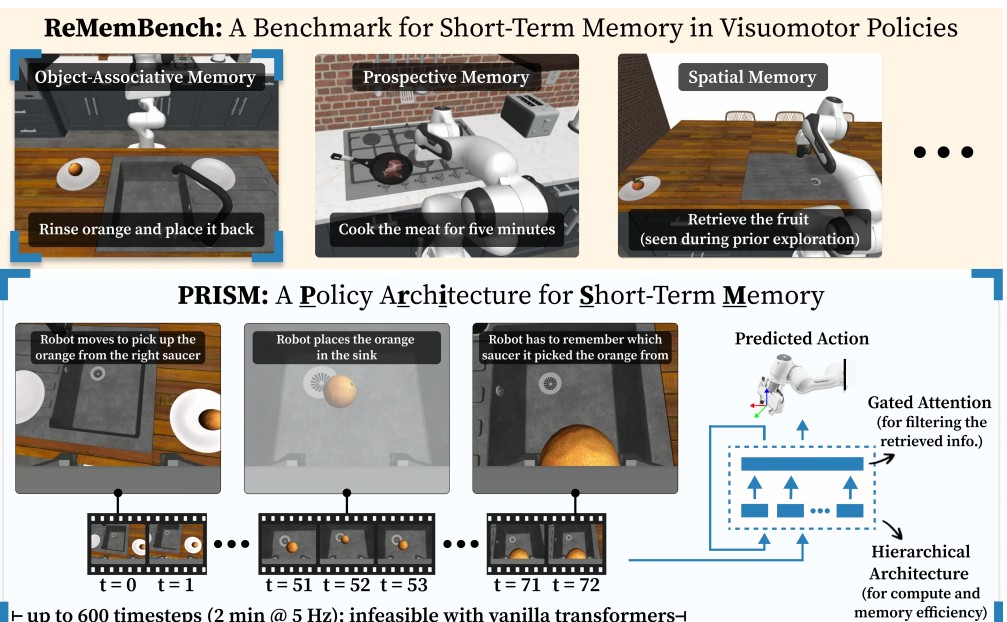

Figure 1: We propose REMEMBENCH, a benchmark for evaluating visuomotor policies with short-term memory (top), and PRISM, an architecture for visuomotor policies with gated attention and a hierarchical architecture, enabling robots to use short-term memory for long-horizon tasks (bottom). To bridge this gap, we introduce **PRISM**: a **P**olicy **A**rch**i**tecture with **S**hort-Term **M**emory for visuomotor control. PRISM maps sequences of past sensory inputs and actions—serving as short-term memory—to future actions in a closed-loop setting. PRISM extends vanilla transformer architecture with two key modifications. First, a gated attention mechanism selectively filters information retrieved from memory, suppressing irrelevant information Qiu et al. (2025). Second, a hierarchical design compresses recent history into summary tokens and then models long-horizon dependencies by attending over this compact token set rather than the full sequence, substantially reducing computational cost. By combining filtering of irrelevant information with a resource-efficient hierarchical architecture, PRISM scales short-term memory to sequences of up to two minutes at 5 Hz, an order of magnitude larger than prior approaches.

To systematically evaluate visuomotor policies with short-term memory, we introduce **REMEM-BENCH**, a benchmark inspired by short-term memory categories in cognitive science ( Figure 1). Existing embodied AI benchmarks (Pasukonis et al., 2022; Fang et al., 2025) either probe memory indirectly or focus narrowly on a single aspect, such as spatial memory, making it difficult to assess memory capabilities holistically. REMEMBENCH instead spans four categories of short-term memory—spatial, prospective, object-associative, and object-set—across eight visuomotor tasks. By unifying these categories within a single benchmark, REMEMBENCH encourages development of general memory mechanisms, rather than siloed task-specific solutions.

PRISM outperforms state-of-the-art methods, including a recent transformer-based visuomotor policy that adds an auxiliary *past-token prediction* loss to encourage short-term memory use (Torne et al., 2025), recurrent models (Vennerød et al., 2021; Gu & Dao, 2023), and short-term memory management strategies adapted from large language models (OpenAI, 2025), across both simulation and real-world tasks. On REMEMBENCH, PRISM achieves an absolute gain of 11% of the strongest baseline aggregated across tasks. Its performance scales with memory length: training with longer short-term memory windows (1→512 timesteps) yields progressively higher success rates. Moreover, by augmenting the state with recent observations, PRISM resolves action ambiguity under visually identical inputs (*e.g.*, "just grasped" vs. "about to place"), reducing multimodality and improving performance on benchmarks not explicitly designed to test memory. In particular, it yields 14-point gains on RoboCasa (Nasiriany et al., 2024) and +11-point gain on LIBERO (Liu et al., 2023) over its no-memory variant. In a real-world adaptation of a ReMemBench task, PRISM doubles the success rate (30% vs. 15% for the strongest baseline), while a policy without mem-

ory achieves 0%. Together, these results establish PRISM as an effective and scalable approach to memory-augmented visuomotor control across both benchmarks and real-world tasks.

## 2 RELATED WORK

**Short-Term Memory in Machine Learning.** Recurrent architectures such as RNNs (Elman, 1990), LSTMs (Hochreiter & Schmidhuber, 1997), and more recent state-space models like Mamba (Gu & Dao, 2023) have been widely used to capture temporal dependencies. Their memory capacity, however, is tied to the parameter count, and information is only indirectly accessible through hidden states, which hinders scalability and leads to the credit assignment problem in back-propagation through time. Transformers (Vaswani et al., 2017) address these limitations by decoupling memory capacity from parameters and enabling direct access to stored representations. Yet, they introduce new challenges: attention scales quadratically with context length, visual inputs yield high token costs (often $10^2$ per image), and irrelevant tokens can degrade performance (Hong et al., 2025). Stability and efficiency have been pursued through attention sinks (Xiao et al., 2023), gated attention (Qiu et al., 2025), hierarchical attention Yang et al. (2016), hybrid attention mechanism with global and sliding window Beltagy et al. (2020), hybrid attention with segment-level recurrence Dai et al. (2019); Parisotto et al. (2019), but these remain largely unexplored in closed-loop visuomotor control.

**Short-Term Memory in Visuomotor Imitation Learning.** In visuomotor imitation learning, short-term memory has traditionally been handled by recurrent policies such as LSTMs Abolghasemi et al. (2019); Mandlekar et al. (2021), but these models struggle with long horizons due to credit assignment in back propagation through time Bengio & Frasconi (1993). Recent transformer-based visuomotor policies largely discard history altogether, conditioning only on the current frame or a short window (Chi et al., 2023; Zheng et al., 2024; Black et al., 2024; Bjorck et al., 2025; Team et al., 2025), or, conversely, by aggressively compressing entire interactions into a few embedding vectors to reduce computation, which removes fine-grained temporal and spatial information Fang et al. (2019). Recent work addresses this limitation with explicit memory mechanisms to transformer-based policies. External memory modules augment policies with structured stores of past states, such as the spatial memory module in SAM2Act+ (Fang et al., 2025) or the perceptual–cognitive memory bank in MemoryVLA (Shi et al., 2025). Auxiliary objectives provide another route, as in Past-Token Prediction (PTP) (Torne et al., 2025), which reconstructs past tokens to regularize policies. In contrast, our work explores an alternative: integrating memory directly into the transformer backbone through gated attention and hierarchical architecture, enabling selective retention and efficient scaling without external modules or auxiliary losses.

**Memory Benchmarks.** Several embodied benchmarks involve partial observability, where memory is useful but not isolated. ALFRED (Shridhar et al., 2020), Habitat MultiON (Wani et al., 2020), and FindingDory (Yadav et al., 2025) ask agents to recall object states or past goals, but performance also depends heavily on perception, language, and exploration. In contrast, recent work introduces

| BENCHMARK | DOMAIN | SHORT-TERM MEMORY CATEGORIES | | | |
|---|---|---|---|---|---|
| | | SPATIAL | PROSPECTIVE | OBJECT-ASSOCIATIVE | OBJECT-SET |
| ALFRED (Shridhar et al., 2020) | Household manipulation | ✓ | ✗ | ✗ | ✗ |
| Habitat MultiON (Wani et al., 2020) | Navigation | ✓ | ✗ | ✗ | ✗ |
| FindingDory (Yadav et al., 2025) | Navigation + manipulation | ✓ | ✓ | ✗ | ✗ |
| Memory Maze (Pasukonis et al., 2022) | Navigation | ✓ | ✗ | ✗ | ✗ |
| Memory Gym (Pleines et al., 2023) | Synthetic gridworlds | ✓ | ✗ | ✓ | ✗ |
| POPGym (Morad et al., 2023) | Synthetic games | ✓ | ✗ | ✓ | ✗ |
| MemoryBench (Fang et al., 2025) | Manipulation | ✓ | ✗ | ✓ | ✗ |
| Mikasa-Robo (Cherepanov et al., 2025) | Manipulation | ✓ | ✓ | ✗ | ✗ |
| **REMEMBENCH** | Navigation + manipulation | ✓ | ✓ | ✓ | ✓ |

Table 1: Comparison of memory-focused embodied AI benchmarks. Rows highlighted in blue explicitly require memory by design (i.e., identical observations can correspond to different actions depending on history). Rows in orange involve memory implicitly but do not explicitly test for it. REMEMBENCH uniquely covers all short-term categories in visuomotor tasks.

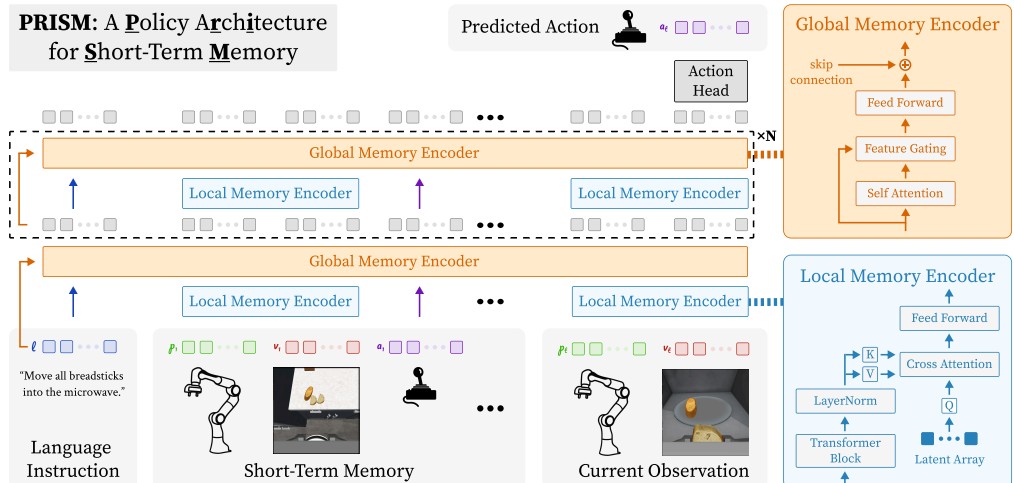

Figure 2: PRISM applies gated attention to filter historical context and hierarchical summarization to scale attention over long interaction histories, improving causal transformer policies trained with behavior cloning by improving robustness to noisy histories and reducing computation.

benchmarks that explicitly enforce memory dependence. Memory Maze (Pasukonis et al., 2022) evaluates spatial recall in procedurally generated mazes. Memory Gym (Pleines et al., 2023) and POPGym (Morad et al., 2023) provide gridworld and game-based tasks targeting short-term recall. MemoryBench (Fang et al., 2025) defines manipulation tasks with identical observations yield different actions depending on history, but mainly targets a single spatial memory type where specialized methods already reach near-saturated performance. Mikasa-Robo (Cherepanov et al., 2025) instead studies reinforcement learning algorithms with memory for relatively short-horizon manipulation (episodes up to ∼180 steps). In contrast, ReMemBench, with its human teleoperation dataset, is designed to study imitation learning under partial observability. It spans both navigation and manipulation with longer-horizon, multi-subtask episodes (up to ∼2.8k steps), organized into four memory categories: spatial, object-set, object-associative, and prospective.

In summary Table 1, REMEMBENCH encompasses short-term memory of various categories, encouraging unified memory architectures that generalize across tasks.

## 3 PRISM: A POLICY ARCHITECTURE FOR SHORT-TERM MEMORY

### 3.1 FORMULATION, BACKGROUND, AND CHALLENGES

**Problem Formulation.** We study imitation learning under partial observability, where the robot receives observations $o_t$ (*e.g.*, RGB images and proprioception) and produces actions $a_t$ in a closed loop. The training dataset $\mathcal{D}_{\text{train}} = \{(\tau^i, l^i)\}_{i=1}^{N}$ consists of $N$ expert trajectories $\tau^i = \{(o_t^i, a_t^i)\}_{t=1}^{T^i}$, paired with a task language instruction $l^i$. We learn a policy $\pi_\theta$ via behavior cloning, conditioning on the interaction history up to time $t$:

$$a_t \sim \pi_\theta(a \mid o_{\leq t}, a_{<t}, l).$$

**Training Objective.** We optimize a behavior cloning loss to match the expert distribution,

$$\mathcal{L}(\theta) = - \mathop{\mathbb{E}}_{(\tau,l) \sim \mathcal{D}_{\text{train}}} \sum_{t=1}^{T(\tau)} \log p_\theta(a_t \mid o_{\leq t}, a_{<t}, l).$$

where $p_\theta(\cdot)$ denotes a parametric action distribution. We employ action chunking, predicting the next $l$ future actions at each timestep (Zhao et al., 2023).

**Evaluation.** During inference, the policy predicts actions autoregressively with language tokens forming a static prefix, while observation and action tokens accumulate over time. Once the horizon budget $n$ is reached, only the most recent $n$ steps are retained in memory. The learned policy is evaluated on its ability to generalize to novel initializations and object placements.

**Model Architecture.** At each timestep $t$, the policy processes a truncated history

$$H_t = (o_k, a_k, o_{k+1}, a_{k+1}, \ldots, o_{t-1}, a_{t-1}, o_t), \quad k = \max(t - n, 1),$$

together with the task description $l$, where $n$ is a fixed memory budget on the number of retained interaction steps. Each observation $o_t = (v_t, p_t)$ consists of an image $v_t$ and a proprioceptive state $p_t$. Images are encoded into $k_v$ visual tokens using an image encoder, proprioceptive states into $k_p$ tokens, and past actions into $k_a$ tokens via an action embedding network. The instruction $l$ is encoded with a pretrained text encoder into $k_l$ language tokens.

All tokens are packed in temporal order with a causal mask, and positional encodings preserve temporal and (for images) spatial structure. The resulting token sequence is

$$[l]_{1:k_l}, \ [p_k]_{1:k_p}, [v_k]_{1:k_v}, [a_k]_{1:k_a}, \ [p_{k+1}]_{1:k_p}, [v_{k+1}]_{1:k_v}, [a_{k+1}]_{1:k_a}, \ \ldots, \ [p_t]_{1:k_p}, [v_t]_{1:k_v}.$$

This sequence is processed by a transformer-based backbone (Vaswani et al., 2017). The transformer maintains the sequence within its context window, which serves as the short-term memory. At each step, the hidden state of the last observation token $[v_t]_{k_v}$ is passed through an action head to produce the parameters of $p_\theta(a_t \mid \cdot)$.

**Challenges.** Attending to long histories is crucial for tasks that require memory but raises two key challenges. First, long histories often include irrelevant information that induces spurious correlations during training. Second, naively attending to all past tokens leads to prohibitive memory and computation costs during both training and evaluation.

## 3.2 SELECTING RELEVANT FEATURES

Transformers retain a record of recent inputs through their context window, effectively forming the short-term memory of a policy. However, long input histories can also introduce irrelevant information, leading models to learn spurious correlations (De Haan et al., 2019; Wen et al., 2020), *e.g.*, associating the positions of past distractor objects with a decision to place an object on the left or right. At test time, such correlations often fail to generalize, resulting in poor performance. This issue is especially pronounced in behavior cloning, where even a small initial error $\epsilon$ can compound over time as $O(T^2\epsilon)$ (Ross et al., 2011; Spencer et al., 2021) for horizon length $T$. Thus, while memory is essential for long-horizon household tasks, it also introduces noise that makes learning more difficult.

Transformers with longer context windows are particularly vulnerable to this issue due to the nature of the attention mechanism: the softmax operation distributes normalized weights throughout the history, potentially amplifying the influence of irrelevant tokens (Hong et al., 2025).

To mitigate this, we use a gating mechanism that regulates the information flow after the attention. Specifically, we modulate the output of each attention block using a learnable gate conditioned on the current input features. Conditioning the gate on the input features allows the model to adaptively control which retrieved information to suppresses as the interaction progresses. Let $x$ denote the input to a transformer block, and let $z = \text{Attn}(\text{Norm}(x))$ denote the attention output. We compute the gated output as:

$$\tilde{z} = g(x) \odot z,$$

where $g : \mathbb{R}^d \to (0,1)^d$ is a gating function, implemented as an MLP with sigmoid activation, and $\odot$ denotes element-wise multiplication. The gated output is then incorporated via the standard residual connection: $y = x + \tilde{z}$, followed by the feed-forward layer (See Figure 2, top-right).

## 3.3 REDUCING MEMORY AND COMPUTE FOOTPRINT

Naively attending over all tokens from $n$ timesteps in history, each with $k_v$ visual tokens, $k_p$ proprioceptive tokens, and $k_a$ action tokens, incurs a quadratic compute and memory cost during training:

$$\mathcal{O}\big((nk_{\text{tot}})^2\big), \text{ where } k_{\text{tot}} = k_p + k_v + k_a$$

Similarly, during evaluation, it incurs a linear cost per step, $\mathcal{O}(nk_{\text{tot}})$, assuming key-value caching. However, as the context grows, even linear compute and memory costs can become prohibitive.

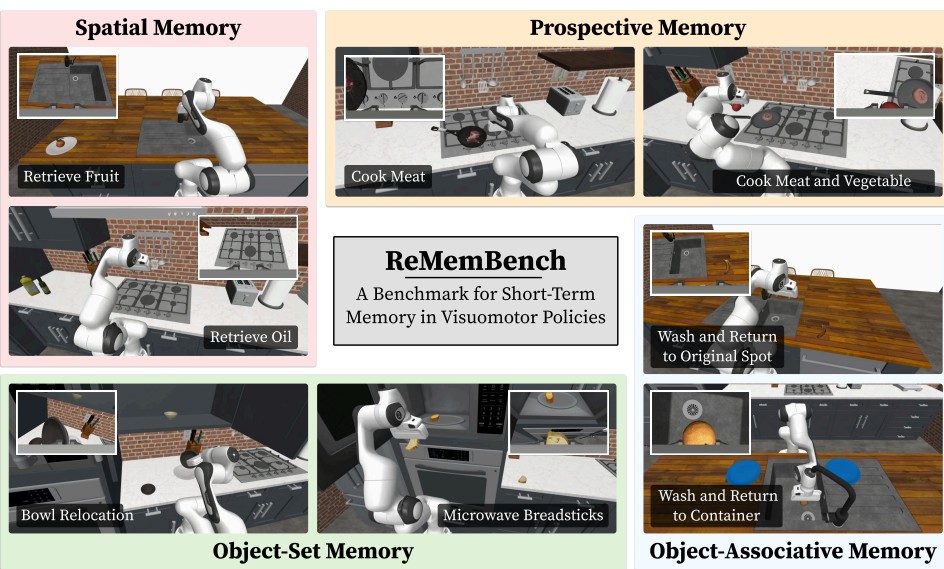

Figure 3: Guided by the cognitive science literature, ReMemBench decomposes short-term memory into several functional categories, such as Spatial, Prospective, Object-Set, and Object-Associative.

To improve scalability, we propose a hierarchical architecture that first splits the sequence into a local encoding of memory, then aggregates the compressed outputs using a global memory encoder. This approach avoids processing the entire token set at once and splits the computation into two parts (Howard et al., 2017). If each local window compresses $k_{\text{tot}}$ tokens down to $k_{\text{tot}}/m$, then the total compute and memory cost during training becomes ( Appendix A):

$$O\big(nmk_{\text{tot}} + (nk_{\text{tot}}/m)^2\big)$$

which is approximately $m^2\times$ cheaper than original $\mathcal{O}((nk_{\text{tot}})^2)$ when $m \ll n$. Similarly, during evaluation, the hierarchical architecture yields a factor-$m$ gain in memory and compute: attention is computed over only $nk_{\text{tot}}/m$ tokens instead of $nk_{\text{tot}}$, and the number of cached key-value tokens is likewise reduced by $m$.

Concretely, at each step, a *local memory encoder* first processes the current timestep's sensory information using a transformer block, then uses a perceiver-style resampler (Jaegle et al., 2021) to compress $k_{\text{tot}} \rightarrow k_{\text{tot}}/m$ ( Figure 2, bottom-right). A *global memory encoder* then computes attention over the concatenated compressed tokens across time ( Figure 2, top-right). The resulting token with global information is *broadcasted* back to the original $k_{\text{tot}}$ tokens and fused via a residual skip from the original inputs to the local memory encoder. The skip connection helps preserve per-token individuality while injecting global information.

**Summary.** PRISM combines (i) gated attention, which adaptively filters irrelevant information from memory, and (ii) a hierarchical architecture, which reduces the compute and memory costs of incorporating long histories. Together, they improve robustness to spurious correlations while allowing the policy to scale memory to longer horizons efficiently. Both components integrate seamlessly with a causal transformer policy trained using a standard behavior cloning objective; no auxiliary losses or extra supervision are required..

## 4 ReMemBench: A Benchmark for Short-Term Memory in Visuomotor Policies

ReMemBench is designed to evaluate short-term memory in visuomotor policies. Guided by the cognitive science literature, we decompose short-term memory into several functional categories (Figure 3). Diversity in categories promotes developments in general memory mechanisms and not custom, non-generalizable solutions for a particular task. Each category is instantiated with two household-manipulation tasks. All tasks are framed as imitation learning problems under partial

observability, ensuring that immediate sensory inputs are insufficient and success requires within-episode recall.

**Spatial Memory.** The ability to recall object locations (O'keefe & Nadel, 1978). We created the following tasks, taking inspiration from the "Belongings: Delayed Recall" subtest of the Rivermead Behavioural Memory Test (Wilson et al., 1985):

- *Retrieve Fruit:* A fruit is placed somewhere in the kitchen and is out of view from a neutral start. The robot must remember its location, fetch it, and place it in the sink. Fruit type and placement differ across demonstrations and novel fruit positions at test time.
- *Retrieve Oil:* Several household items are arranged and not visible from the start. The robot must find the specified oil bottle and pick it up. The bottle instance and location vary across demonstrations with novel positions at test time.

**Prospective Memory.** The ability to retain intentions over a delay and execute them at the right time (Einstein & McDaniel, 1990). We consider the following tasks, inspired by the Prospective-Retrospective Memory Questionnaire (Smith et al., 2000):

- *Cook Meat:* A pan with meat sits on the stove. The robot must turn the stove on, wait the required duration, then turn it off. The target duration, pan type, and meat type vary by demonstration.
- *Cook Meat and Vegetable:* Meat begins on the stove, and a vegetable is nearby. The robot must turn the stove on, add the vegetable after a specified duration, and turn the stove off after another specified duration so both items meet their cooking times. Durations, pan type, ingredient types, and vegetable placement vary.

**Object-Associative Memory.** The ability to recall object-location associations (Mayes et al., 2007). We propose the following, inspired by the Paired Associates Learning test (Sahakian et al., 1988):

- *Wash and Return to Container:* Two saucers sit to the left and right of the sink; one holds a fruit. The robot must wash the fruit and return it to the *same* saucer. Fruit and saucer types and the saucer side vary, with novel placements at test time.
- *Wash and Return to Original Spot:* A fruit starts on the countertop. The robot must move it to the sink to wash it, and place it back at its original location (within a small window). Initial positions, fruit, and saucer type vary across demonstrations with novel test-time positions.

**Object-Set Memory.** The ability to maintain and update sets of multiple objects across time (Luck & Vogel, 1997; Makovski & Jiang, 2009). We consider the following tasks, inspired by object set maintenance paradigms in working memory, particularly the Counting Span task (Case et al., 1982):

- *Microwave Breadsticks:* A plate holds multiple breadsticks, and the microwave is initially out of view. The robot must move all breadsticks into the microwave and close the door, keeping track of how many remain. Counts, types of bread, and placements vary with novel test-time positions.
- *Relocate Bowls:* Bowls with distractor plates nearby sit beside a cabinet. The robot must transfer all and only the bowls into the cabinet while tracking the remaining count. Bowl types, counts, and placements vary with novel test-time positions.

REMEMBENCH builds on assets from the ROBOCASA environment (Nasiriany et al., 2024), with each task provided with 50 expert demonstrations for training. Policies are trained via imitation learning and evaluated by success rate. The benchmark and data will be fully open-source.

## 5 EXPERIMENTS

**Baselines.** We group our baselines into four categories according to their memory mechanisms and application domains and evaluate them on REMEMBENCH. **Recurrent baselines** include LSTM (Vennerød et al., 2021) and Mamba (Gu & Dao, 2023), which compress history into hidden states. **Segment-level recurrent transformers** include Transformer-XL (TrXL) (Dai et al., 2019) and Gated Transformer-XL (GTrXL) (Parisotto et al., 2019), which combine recurrence with attention. We further include a **linear-attention transformer** (Katharopoulos et al., 2020), which replaces softmax with a linear kernel to obtain long-context attention with reduced compute and memory. Finally, under **visuomotor policy architectures for imitation under partial observability**, we consider Past Token Prediction (PTP) (Torne et al., 2025), which adds an auxiliary loss to predict past actions, the Scene Memory Transformer (SMT) (Fang et al., 2019), which compresses entire trajectories into scene-level memories via attention pooling, and SAM2Act++ (Fang et al.,

Figure 4: (Left) Successful real-world rollout of PRISM on *Wash and Return to Container*. (Right) Without memory, the policy cannot disambiguate between 'just picked up' vs. 'about to place.'

2025) that creates a compact state representation by storing the pixel co-ordinates of objects observed in the past along with the current observation.

**Q1.** How does PRISM compare to state-of-the-art methods on memory-demanding tasks?

**REMEMBENCH Results.** Recurrent models such as LSTM and Mamba compress the entire history into a single hidden state, limiting capacity and struggles with assigning credits to the correct timestep using backpropogation through time: in a "turn off the stove after a delay" task, the learning signal from the decision should backpropagate through hundreds–thousands of updates, so gradients vanish or get overwritten. Consequently, LSTM and Mamba achieve only 9% and 13% success versus PRISM's 45% (drops of 36 and 32 points) (Table 2). Segment-level recurrent transformers like TrXL and GTrXL still rely on recurrence, so long-horizon credit assignment remains challenging, yielding 15% and 26% (−30 and −19 points) (Bengio & Frasconi, 1993). Linear-Attention uses effectively compresses the entire history into a single matrix; this representation suffers from forgetting over long horizons and reaches only 22% (−23 points) (He & Garner, 2025).

Prior transformer-based methods also struggle to regulate the retrieved information. PTP improves recall via an auxiliary loss but does not filter irrelevant information, leading to only 15% success (−30 points). Scene-level memory models like SMT compress entire trajectories into a pooled embedding, discarding fine-grained temporal and spatial information, reaching 36% (−9 points). In contrast, PRISM, with token-level features, combines attention and feature-wise gating to retrieve information and suppress irrelevant information. Lastly, SAM2Act++ focuses solely on the spatial coordinates of objects in memory, but due to a lack of semantics information, it achieves only 19% success (−11 points) across all tasks in REMEMBENCH.

**Real-World Results.** We evaluate PRISM on a real-world adaptation of the 'Wash and Return to Container' task from REMEMBENCH (Figure 4). Without access to memory, the baseline policy fails completely. The transformer-based PTP baseline achieves a 0.15 success rate, while PRISM doubles this, reaching 0.30. This substantial improvement demonstrates PRISM's effectiveness in real-world tasks, where temporally extended short-term memory is critical.

**Takeaway 1.** PRISM achieves twice the success rate of state-of-the-art methods on REMEMBENCH and real world.

**Q2.** Does PRISM provide benefits on standard benchmarks that do not explicitly test memory?

Expanding the state space with recent observations provides temporal information often missing from the current state alone. In ROBOCASA, for instance, the same visual input may require different actions depending on whether an object was just grasped or is about to be placed. Short-term memory helps disambiguate such situations by reducing multimodality in the state-conditioned action

Table 2: Success rate across categories on REMEMBENCH.

| Group | Method | Spatial | Prospective | Obj-Assoc. | Obj-Set | Avg |
|---|---|---|---|---|---|---|
| REMEMBENCH | LSTM (Vennerød et al., 2021) | 0.14 | 0.04 | 0.12 | 0.04 | 0.09 |
| | Mamba (Gu & Dao, 2023) | 0.04 | 0.34 | 0.07 | 0.08 | 0.13 |
| | TrXL (Dai et al., 2019) | 0.06 | 0.28 | 0.12 | 0.12 | 0.15 |
| | GTrXL (Parisotto et al., 2019) | 0.20 | 0.30 | 0.30 | 0.24 | 0.26 |
| | Linear-Attention (Katharopoulos et al., 2020) | 0.10 | 0.40 | 0.20 | 0.18 | 0.22 |
| (Four Tasks) | Standard Transformer (Vaswani et al., 2017) | 0.12 | 0.44 | 0.20 | 0.20 | 0.24 |
| | PTP (Torne et al., 2025) | 0.14 | 0.34 | 0.06 | 0.06 | 0.15 |
| | SMT (Fang et al., 2019) | 0.50 | 0.40 | 0.26 | 0.28 | 0.36 |
| | PRISM | 0.40 | 0.80 | 0.25 | 0.35 | 0.45 |
| REMEMBENCH (ALL) | PTP (Torne et al., 2025) | 0.13 | 0.24 | 0.08 | 0.17 | 0.16 |
| | SAM2Act++ (Fang et al., 2025) | 0.14 | 0.08 | 0.24 | 0.30 | 0.19 |
| | PRISM | 0.37 | 0.35 | 0.15 | 0.33 | 0.30 |

distribution, making the problem tractable. Empirically, adding memory over the past $64$ timesteps improves success rate from $0.33$ to $0.47$, while extending it to $256$ timesteps slightly degrades performance, likely due to increased accumulated noise ( Figure 5, Left). Full comparisons on RoboCasa and LIBERO Liu et al. (2023) are provided in Appendix C.5

> **Takeaway 2.** PRISM boosts performance by $14$ percentage points on atomic ROBOCASA tasks that do not explicitly require short-term memory.

> **Q3.** How does PRISM's performance scale with increasing memory capacity?

PRISM exhibits strong scalability with increasing memory size on REMEMBENCH ( Figure 5, Right). As the memory window expands from $n = 1$ to $n = 512$, the success rate increases by $0.26$, indicating that PRISM continues to extract useful information from temporally extended memory without saturation. This trend highlights PRISM's suitability for tasks that demand reasoning over extended temporal information.

> **Takeaway 3.** Increasing the memory capacity from $n = 1$ to $n = 512$ improves success by $26$ percentage points with steady gains and no saturation.

> **Q4.** Is gated attention effective for filtering irrelevant information in visuomotor policies?

The gating mechanism in PRISM is essential for selectively controlling memory usage. Without it, the model suffers a $16$ percentage points performance drop, highlighting the importance of filtering out irrelevant information retrieved from memory ( Figure 5, Middle). We also evaluate a variant that uses a per-layer scalar gate while ignoring per-step input features. Although this offers a modest gain over no gating ($3$ percentage points), it performs significantly worse than PRISM's input feature-aware gating ($-13$ percentage points). Furthermore, we test adding attention sink tokens instead of gating, which allows the model to attend to a dummy token rather than irrelevant information from memory (Xiao et al., 2023). While successful in language models (OpenAI, 2025), attention sinks prove ineffective, highlighting the need for further investigation into different memory filtering mechanisms in visuomotor policies.

> **Takeaway 4.** Gating mechanism filters out irrelevant information yielding a $16$ percentage points improvement in performance on REMEMBENCH tasks.

> **Q5.** What computational and memory efficiencies are gained from the hierarchical architecture?

PRISM scales efficiently over extended temporal sequences because it compresses local information before any global attention, allowing global operations to run on a compact set of summary tokens rather than the full sequence. Moreover, it only needs to cache the summary tokens, rather than all the tokens. At $n = 512$, this yields $0.20\times$ peak GPU memory ($-80\%$) and $0.60\times$ runtime ($-40\%$) relative to both full attention and hybrids with global attention + sliding window attention (Beltagy et al., 2020) (Figure 6). We observe a increase in hybrid method's runtime performance at $n = 4$, likely due to CUDA kernel scheduling rather than algorithmic effects.

> **Takeaway 5.** At longer context lengths, PRISM reduces peak GPU memory by $80\%$ and runtime by $40\%$ vs. full and hybrid attention during evaluation.

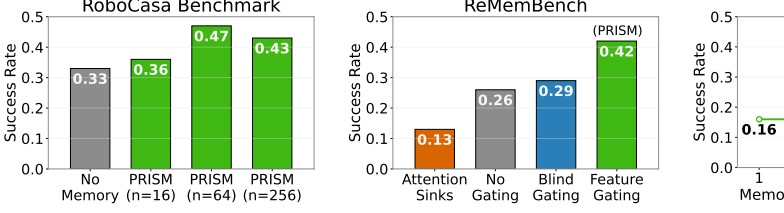

Figure 5: *Left:* PRISM demonstrates improvement on RoboCasa benchmark by adding short-term memory. *Middle:* PRISM's gating mechanism effectively filters out irrelevant information for memory-intensive tasks. *Right:* PRISM's performance scales with memory capacity.

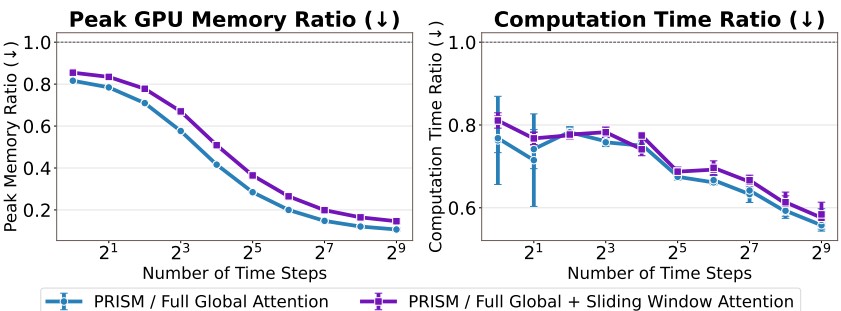

Figure 6: PRISM is both computationally and memory efficient at handling long contexts.

## 6  LIMITATIONS AND CONCLUSION

**Benchmark headroom.** Absolute success rates on REMEMBENCH cluster around the 30% range. We view this not as a weakness of memory, but as evidence that the long-horizon tasks are hard. Improving overall performance is essential to demonstrate the benefits of history and encourage co-improvements in perception, control, not only in the memory-related module itself.

**Short vs. long-term memory.** In this work, we focus on *short-term* memory as retention of a recent, finite window of sensory–motor history (on the order of the last few minutes) Atkinson & Shiffrin (1968), which the policy uses to predict actions for control. Looking forward, we envision augmenting PRISM with a persistent *long-term* memory that accumulates information over hours, days, or even the lifetime of the system (e.g., via replay-like buffers or slowly updated parameters) and retrieves relevant traces back into short-term memory when needed for decision-making, enabling lifelong learning in robots.

**Internet-scale priors.** Learning to *use* short-term memory effectively can be data-intensive. Our study operates at a scale far smaller than internet-sized corpora. An exciting direction is to pretrain memory modules on large, internet-scaled videos and study how such priors transfer to visuomotor control for short-term memory use.

We introduce PRISM, a visuomotor policy architecture that incorporates short-term memory through gated attention and a hierarchical design. To systematically evaluate memory in visuomotor control, we propose REMEMBENCH with eight tasks spanning different categories: spatial, prospective, object-associative, and object-set. PRISM outperforms state-of-the-art baselines across (1) all REMEMBENCH tasks, (2) ROBOCASA atomic tasks not explicitly designed to test memory, and (3) real-world evaluations. Its performance also scales with memory capacity, with longer histories yielding consistent gains. Together, PRISM and REMEMBENCH provide a foundation for developing and evaluating memory-augmented visuomotor policies for long-horizon tasks.

## 7  DISCUSSION

In this work, we represent short-term memory using recent sensory-motor experience. In principle, hand-engineered state representations could reduce reliance on explicit memory for specific aspects of a task. However, even with ideal indicator functions for some state variables, robots would still require memory for other critical dependencies. For example, in *Wash and Return to Container*, a perfect wetness indicator could distinguish "just washed" from "about to wash," but the policy must still remember which container the object originally came from and where that container was placed. Similarly, in *Wash and Return to Same Location*, the robot must recall the original pick-up location on the table, which is no longer directly observable once the scene changes.

More broadly, replacing memory with hand-engineered state would require an ever-growing set of task-specific indicators, spanning fine-grained properties (*e.g.*, container identity, pose, wetness, time since last interaction, etc) as well as higher-level variables (*e.g.*, task progress and outcomes of prior attempts). Designing and maintaining such features across diverse objects, layouts, and tasks for general-purpose robots may become intractable. Taken together, these considerations underscore the importance of using general representations of memory, rather than relying on an expanding set of hand-crafted state variables.

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

## A MEMORY AND COMPUTE FOOTPRINT

**Training.** Assuming each local encoder compresses $k_{\text{tot}}$ tokens to $k_{\text{tot}}/m$, there are $n$ local blocks, each processing $k_{\text{tot}}$ tokens. The local self-attention per block costs $\mathcal{O}(m^2)$ and is invoked $n\, k_{\text{tot}}/m$ times, yielding a total local compute of $\mathcal{O}(n\, m\, k_{\text{tot}})$. After compression, the global module attends over $n\, k_{\text{tot}}/m$ summary tokens, incurring $\mathcal{O}\big((n\, k_{\text{tot}}/m)^2\big)$. Thus, the overall training-time compute decomposes into a linear-in-$k_{\text{tot}}$ local term plus a quadratic-in-the-compressed-length global term:

$$\underbrace{\mathcal{O}(n\, m\, k_{\text{tot}})}_{\text{local}} + \underbrace{\mathcal{O}\big((n\, k_{\text{tot}}/m)^2\big)}_{\text{global}}.$$

Training-time *memory* can also benefit from the reduced global length (lesser activation memory), but these savings are less pronounced in practice because modern optimizations (e.g., FlashAttention tiling/splitting) already reduce activation memory, the major contributor to training-time footprint for long-sequence transformers.

**Evaluation.** The same decomposition applies to evaluation compute, but the memory benefit is much clearer. Operating global attention over $n\, k_{\text{tot}}/m$ summary tokens instead of all $n\, k_{\text{tot}}$ tokens reduces both global-attention compute and the key-value (KV) cache by a factor of $\approx m$. Concretely, at inference, the model only attends to $n\, k_{\text{tot}}/m$ tokens and stores proportionally fewer KV entries, yielding an $\approx m\times$ reduction in global-attention runtime and memory relative to full/global attention over the uncompressed sequence. Empirical evaluation-time measurements (compute and memory) are reported in the main paper.

**Baselines** To quantify empirical compute and memory savings, we compare PRISM to two baselines: (i) standard full attention over the entire history, and (ii) a hybrid with full global attention plus a sliding window. For each method, we run a unit block 20 times, recording wall-clock time and peak memory. We then normalize these measurements by the number of learnable parameters and report PRISM-to-baseline ratios for both compute time and memory.

## B    Implementation Details

**Baseline Details**    *PTP:* Predicts the past $k$ actions jointly with $l$ future actions; we follow the original hyperparameters and set $k=16$ (other settings as reported in the paper).
*LSTM:* We use the standard PyTorch implementation, match the number of stacked layers to PRISM's Transformer depth, and increase the hidden size so the total trainable parameters closely match PRISM.
*Mamba:* We use the `mampy` implementation, again matching the number of layers to PRISM and scaling the hidden size to approximate parameter parity.

**PRISM Architecture Details**    We set $k_p=1$, $k_v=198 \times 2$ (tokens per camera $\times$ cameras), and $k_a=1$. In all experiments, the *local* encoder compresses the $(k_p+k_v)$ tokens to a single summary token; thus the per-step token count changes from $(k_p+k_v+k_a)=398$ to $(1+k_a)=2$, yielding a compression factor $m = \frac{k_p+k_v+k_a}{1+k_a} = \frac{398}{2} = 199$. We instantiate the Transformer backbone with a standard LLaMA-style implementation. Visual features are extracted using *CrossMAE* (Fu et al., 2024) and kept *frozen* during policy training. The action head is a two-layer MLP.

**Data Collection**    For each of the REMEMBENCH tasks, 50 expert demonstrations are collected by a human teleoperator using a spacemouse for imitation learning. Similarly, for the real-world tasks, we collect 100 expert demonstrations using the spacemouse.

**Training Details**    We train for all eight REMEMBENCH tasks using $\sim$384 H100 GPU-hours, and for four ROBOCASA atomic tasks using $\sim$192 H100 GPU-hours.

Table 3: Training hyperparameters.

| Config | Value |
|---|---|
| Optimizer | AdamW |
| Base learning rate | $5 \times 10^{-4}$ |
| Effective batch size | 256 |
| Weight decay | 0.01 |
| Warmup epochs | 2 |
| Total epochs | 200 |
| Action prediction horizon | 32 |
| Proprioception noise (std) | 0.005 |
| Brightness augmentation | Uniform$(-0.1, 0.1)$ |
| Contrast augmentation | Uniform$(0.8, 1.2)$ |
| Random patch masking | 0–16 patches, area $\sim$ Uniform$(1\%, 4\%)$ |
| Number of cameras | 2 |

## C    Additional Analysis

### C.1    Effect of Attention Scale Factor (Sharpness)

| $V = \text{scale\_factor} \cdot \sqrt{d_{\text{head}}}$ | 0.125 | 0.25 | 0.5 | 1.0 (PRISM) | 2.0 | 4.0 | 8.0 |
|---|---|---|---|---|---|---|---|
| Retrieve Fruit | | 0.22 | 0.54 | 0.54 | 0.64 | 0.64 | 0.54 | 0.62 |

Table 4: Effect of attention sharpness on PRISM performance. We scale attention logits by $\frac{V}{\sqrt{d_{\text{head}}}}$ before softmax; PRISM uses the default setting 1.0.

We study the effect of shaping the attention distribution affects performance by scaling the attention logits. Making attention too flat (factor = 0.125) severely hurts success (0.22 vs. 0.64, a $\sim$42% drop), indicating that overly permissive information flow leads to noisy retrieval. In contrast,

moderate increases in sharpness around the PRISM setting (factors 2.0–8.0) cause mild degradation (2–10 pp), suggesting that performance is much more sensitive to insufficient selectivity than to conversative selection. Overall, these results indicate that controlling information flow is crucial for performance, a function implemented by PRISM's gating module and learned directly from data (Table 4).

## C.2 Effect of Attention Dropping

| Method | Spatial | Prospective | Associative | Object-Set | Average |
|---|---|---|---|---|---|
| PRISM w/o Gating | 0.08 | 0.35 | **0.30** | 0.30 | 0.26 |
| PRISM w/o Gating (Attn Drop) | 0.44 | **0.80** | 0.10 | 0.20 | 0.39 |
| PRISM | 0.40 | **0.80** | 0.25 | 0.35 | 0.45 |
| PRISM (Attn Drop) | **0.60** | 0.64 | **0.30** | **0.40** | **0.49** |

Table 5: Effect of noisy attention (token dropping with probability $p = 0.1$) on success rates across the four ReMemBench categories. While noise improves both gated and ungated models, PRISM with learned gating remains strongest overall.

We also examine whether adding noise to attention can improve robustness. We inject noise by randomly dropping tokens with probability $p = 0.1$ [7]. This boosts PRISM's average success by $+4\%$ ($0.45 \rightarrow 0.49$). Applying the same token-drop strategy to PRISM without gating also helps ($0.26 \rightarrow 0.39$), but the full PRISM architecture remains stronger overall: it outperforms the no-gating variant by $+10\%$ with attention dropping (0.49 vs. 0.39) and $+19\%$ without it (0.45 vs. 0.26). These results suggest that noisy attention alone is helpful but insufficient, and that the learned gating module remains crucial for suppressing irrelevant features from retrieval (Table 5).

## C.3 Effect of Gating Module

| Method | Spatial | Prospective | Associative | Object-Set | Average |
|---|---|---|---|---|---|
| Blind Gating | 0.35 | 0.30 | 0.25 | 0.20 | 0.28 |
| Feature Gating (PRISM) | **0.40** | **0.80** | **0.25** | **0.35** | **0.45** |

Table 6: Blind vs. feature-wise gating. Blind gating learns a single scalar gate per layer with similar non-linearity and parameter count, yet feature-wise gating in PRISM achieves substantially higher average success.

We acknowledge that modifications introduced by $g(x)$, such as additional parameters or non-linearities, could confound the claim that performance gains stem from filtering irrelevant information. To isolate the effect of feature-wise gating from these factors, we compare blind gating (86M parameters, sigmoid non-linearity) with feature-wise gating (90M parameters, sigmoid non-linearity). Blind gating, instead of producing a per-feature gate, learns a single scalar gate per layer and thus matches PRISM in non-linearity while maintaining a similar parameter count (less than $5\%$ extra). The resulting performance gap is substantial: feature-wise gating yields a $+17\%$ improvement in average success, indicating that the gains primarily arise from feature-wise information selection rather than increased capacity or any non-linearity (Table 6).

## C.4 Evaluation on Mikasa-Robo

We also evaluate PRISM on Mikasa-Robo and observe gains on some tasks (e.g., InterceptSlow-v0, RememColor5-v0), but overall performance is comparable to LSTM baselines, unlike on REMEM-BENCH, where transformers clearly outperform LSTMs. We hypothesize this gap stems from differences in task design: Mikasa-Robo tasks have much shorter horizons (180 vs. 2.8k timesteps in REMEMBENCH), contain fewer natural distractors, and concentrate most of the relevant information in the initial frames, making them less diagnostic of robust, long-horizon memory mechanisms. These differences in performance invite a systematic study of the factors that strongly influence short-term memory use and downstream performance (Figure 7).

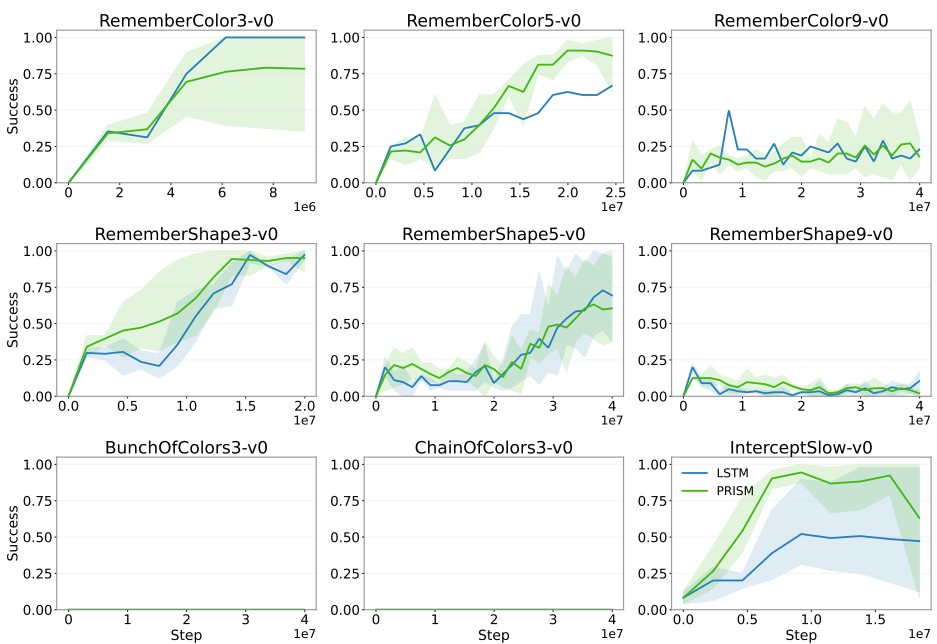

Figure 7: Comparison of PRISM vs. LSTM on Mikasa-Robo.

| Method | Success Rate |
|---|---|
| BC-Transformer | 0.26 |
| Diffusion Policy | 0.26 |
| GR00T-N1-2B | 0.32 |
| PRISM (no memory) | 0.32 |
| PRISM ($n = 256$) | **0.43** |

Table 7: Success rates on RoboCasa. PRISM with memory outperforms both no-memory baselines as well as the pretrained GR00T-N1-2B model.

## C.5 EVALUATION ON LIBERO AND ROBOCASA

We evaluate PRISM (Table 7) on two additional visuomotor benchmarks, RoboCasa and LIBERO with baselines including BC-Transformer, Diffusion Policy, and the large pretrained VLA models like GR00T-N1-2B and OpenVLA. On RoboCasa, PRISM with memory improves over its no-memory variant by an absolute 11 percentage points ($0.32 \rightarrow 0.43$) and also exceeds GR00T-N1-2B by 11 points. On LIBERO (Table 8), PRISM similarly improves over its no-memory counterpart by 12 percentage points averaged over all benchmarks and pretrained OpenVLA finetuned on LIBERO by absolute gain of $16\%$. These results indicate that augmenting policies with short-term memory is beneficial not only for explicitly memory-dependent tasks but also for tasks that appear largely memory-less, by providing additional context and reducing multimodality in the action distribution.

## C.6 ANALYSIS OF THE COMPRESSION FACTOR ($m$)

The compression factor $m$ trades off memory capacity and resource usage. In our experiments, we found $m \approx 200$ (compressing all tokens from the current timestep) to be a practical operating point under a budget of 384 H100 GPU-hours per experiment; for $m \approx 2$, it was difficult to obtain enough gradient steps to reach reasonable performance. Table 9 reports success rates for $m \approx 2, 9, 80$, and 200. As compression becomes more aggressive (larger $m$), we observe a modest but consistent increase in average performance across tasks (from 0.31 at $m \approx 2$ to 0.45 at $m \approx 200$), even though individual categories do not follow a strictly monotonic trend. In contrast, the efficiency gains are

| Method | LIBERO-90 | LIBERO-10 | Object | Spatial | Goal |
|---|---|---|---|---|---|
| OpenVLA Kim et al. (2024) | 0.62 | 0.54 | 0.88 | 0.85 | 0.79 |
| Diffusion Policy Chi et al. (2025) | – | 0.72 | **0.93** | 0.78 | 0.68 |
| PRISM (no memory) | 0.77 | 0.75 | 0.89 | 0.85 | 0.61 |
| PRISM ($n = 256$) | 0.85 | **0.81** | **0.93** | **0.92** | **0.95** |

Table 8: Success rates across the LIBERO benchmark suite. PRISM with memory improves over its no-memory variant and other standard baselines.

| Compression ($m$) | Spatial | Prospective | Associative | Object-Set | Average |
|---|---|---|---|---|---|
| $m \approx 2$ | 0.30 | 0.40 | **0.28** | 0.24 | 0.31 |
| $m \approx 9$ | **0.40** | 0.56 | 0.24 | **0.40** | 0.40 |
| $m \approx 80$ | **0.40** | 0.60 | 0.24 | **0.40** | 0.41 |
| $m \approx 200$ (PRISM) | **0.40** | **0.80** | 0.25 | 0.35 | **0.45** |

Table 9: Effect of compression factor $m$ on success rates across ReMemBench tasks. Larger $m$ corresponds to more aggressive compression.

clear: at sequence length 512, $m \approx 200$ reduces compute by 13% and peak GPU memory by 56% relative to $m \approx 2$ (Figure 8).

# D USE OF LLM

We adhere to the ICLR Code of Ethics and take full responsibility for all content in this paper. We used LLMs only to improve clarity of writing, correcting grammar, refining phrasing, and enhancing readability, and not to generate scientific content, design experiments, analyze data, or draw conclusions.

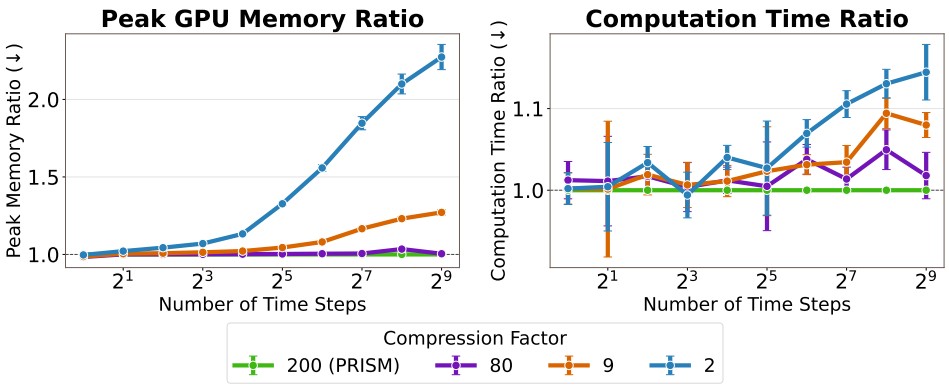

Figure 8: Effect of compression factor $m$ on test-time efficiency. The plot reports peak GPU memory and computation time, both normalized with respect to the $m \approx 200$ (PRISM) configuration, across different sequence lengths.

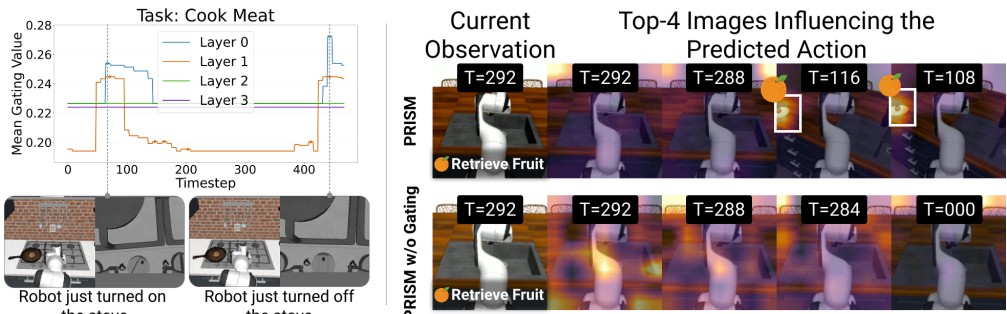

Figure 9: (*Left*) **Mean predicted gating value across timesteps during decision-making.** Higher gating value aligns with intuitive moments in the task, particularly around turning the stove on and off, when memory recall is most critical for disambiguation. A smaller rise near $t \approx 400$ corresponds to the robot initiating the stove turn-off action. We observe a consistent trend in the "Cook Meat" task across successful trials. (*Right*) **Qualitative visualization of the gating effect.** With gating, the policy pays attention to the correct observations and the spatial region where the orange was previously observed from memory. Without gating, attention shifts to task-irrelevant regions, such as the table or sink.

