# OpenReview forum: "Scaling Short-Term Memory of Visuomotor Policies for Long-Horizon Tasks"
_ICLR.cc/2026/Conference — Submitted to ICLR 2026_

### Official Review · Reviewer_1AkL · 2025-10-27

**Soundness:** 2
**Presentation:** 1
**Contribution:** 2
**Rating:** 2
**Confidence:** 4

**Summary:**

This paper addresses the limitation that current visuomotor policies for robotic manipulation are myopic and cannot effectively use short-term memory for long-horizon tasks. The authors introduce PRISM, a transformer-based policy architecture that scales short-term memory through two key innovations: (1) gated attention that selectively filters irrelevant information from memory to prevent spurious correlations, and (2) a hierarchical architecture that compresses local interactions before global attention to reduce computational cost. To systematically evaluate memory capabilities, they also propose REMEMBENCH, a benchmark with 8 household manipulation tasks spanning 4 cognitive memory categories (spatial, prospective, object-associative, and object-set). PRISM achieves 2× the success rate of state-of-the-art baselines on REMEMBENCH and real-world tasks, demonstrating that effective short-term memory mechanisms can significantly improve performance on tasks requiring temporally extended reasoning.

**Strengths:**

- The paper provides an excellent explanation and decomposition of short-term memory into four cognitive categories (spatial, prospective, object-associative, and object-set) in Sec.4, grounded in cognitive science literature. This systematic approach to REMEMBENCH design encourages development of general short-term memory mechanisms rather than task-specific solutions.

- PRISM demonstrates substantial and consistent improvements across multiple evaluation settings. The real-world validation is particularly valuable for demonstrating practical applicability.

**Weaknesses:**

- Insufficient ablation of key hyperparameters. Specifically, no ablation on the compression factor $m$.

- Presentation and formatting issues. The following issues raise concerns about the overall polish and readiness of the submission:
  - Figure-text overlap around lines 422-423 suggests rushed preparation
  - Broken citation for CrossMAE (line 675-676: "*CrossMAE* (?)")


- Unclear future direction on long-term memory: The paper defines long-term memory as "cross-episode retrieval" (line 479-482) but provides insufficient explanation of how this would function in their framework. The distinction between short-term (within-episode) and long-term (cross-episode) memory deserves clearer articulation, particularly regarding whether the authors envision something like retrieval-augmented generation for robotics or continual learning scenarios with replay buffers.

**Questions:**

- The paper categorizes short-term memory into four types (spatial, prospective, object-associative, object-set) based on cognitive science literature. Are these four categories exhaustive, or are they a representative subset of a broader taxonomy? Could you clarify whether REMEMBENCH aims to cover all major categories of short-term memory relevant to robotics, or if additional categories might be added in future work? What criteria from cognitive science guided the selection of these specific four categories?

- Several tasks might potentially be solved without explicit memory through richer state representations. For example, in "Wash and Return to Container" (Fig.4), an observation function that detects wetness could disambiguate "just washed" vs. "needs washing" states, replacing the need to remember the washing action. Similarly, in "Cook Meat," a visual cue indicating stove temperature or cooking duration might substitute for remembering when the stove was turned on. This raises a fundamental question: what is the principled distinction between tasks that inherently require memory vs. tasks solvable through better perception/state engineering or other alternative techniques?

**Details Of Ethics Concerns:**

No ethics review is needed.

---

> ### Author Response · Authors · 2025-11-23
>
> We thank the reviewer for their encouraging feedback, particularly for recognizing the cognitively grounded decomposition of short-term memory in ReMemBench and the value of encouraging general-purpose memory mechanisms. We also appreciate the positive assessment of PRISM’s consistent performance gains and the importance of the real-world validation. Below, we address each of the reviewer’s concerns and provide all the suggested analyses.
>
> > W1: Insufficient ablation of key hyperparameters. Specifically, no ablation on the compression factor
>
> The compression factor allows us to control the resource utilization. In our initial experiments, we found a compression ratio of $m \\approx 200$ to be a practical operating point (by compressing all tokens from the current timestep). Below, we compare performance and normalized compute/memory trade-offs for $m \\approx 80$, $m \\approx 9$, and $m \\approx 2$. Note that with $m \\approx 2$, we found it difficult to obtain enough gradient steps to achieve reasonable performance within our budget of $384$ H100 GPU hours per experiment.
>
> As the compression factor increases (i.e., more aggressive compression), we observe a modest but consistent improvement in average performance across tasks, although per-task results remain noisy and do not exhibit a single monotonic trend. On the other hand, the gains in compute and memory efficiency are consistent and pronounced, as highlighted in Appendix Figure 8 of revised manuscript (or on the website via [this link](https://remembench-prism.github.io/static/images/compression_peak_gpu_time.png)), with $m \\approx 200$ achieving a reduction of $13\\%$ in compute and $56\\%$ in peak GPU memory compared to $m \\approx 2$, measured at a sequence length of $512$.  Details of the calculation of $m$ are provided in Appendix Section B.
>
> | Method | Spatial | Prospective | Associative | Object-Set | Average |
> | :---- | :---- | :---- | :---- | :---- | :---- |
> | $m\\approx2$ | 0.3 | 0.4 | **0.28** | 0.24 | 0.31 |
> | $m\\approx9$ | **0.40** | 0.56 | 0.24 | **0.40** | 0.40 |
> | $m\\approx80$ | **0.40** | 0.60 | 0.24 | **0.40** | 0.41 |
> | $m\\approx200$ (PRISM) | **0.40** | **0.80** | 0.25 | 0.35 | **0.45** |
>
> > W2: Presentation and formatting issues. The following issues raise concerns about the overall polish and readiness of the submission:
> \- Figure-text overlap around lines 422-423 suggests rushed preparation
> \- Broken citation for CrossMAE (line 675-676: "\_CrossMAE\_ (?)")
>
> We thank the reviewer for carefully identifying these presentation issues and apologize for the oversight. The figure-text overlap and the broken CrossMAE citation have been corrected in the revised manuscript.
>
> > W3: Unclear future direction on long-term memory: The paper defines long-term memory as "cross-episode retrieval" (line 479-482) but provides insufficient explanation of how this would function in their framework. The distinction between short-term (within-episode) and long-term (cross-episode) memory deserves clearer articulation, particularly regarding whether the authors envision something like retrieval-augmented generation for robotics or continual learning scenarios with replay buffers.
>
> In our framework, **short-term memory** refers to information retained over a recent, finite window of experience \[1\]. Concretely, we maintain a short-term memory of approximately the past two minutes of sensory–motor history. During evaluation, this memory is reset at the start of each episode.
>
> By contrast, **long-term memory** refers to information that persists over much longer timescales, such as hours, days, or even the lifetime of the system \[1\]. We envision a framework where information from short-term memory can be **consolidated** into a long-term store and later **retrieved** back into short-term memory when relevant. This long-term memory could be implemented as external storage (e.g., replay buffers) and/or embedded in slowly updated model parameters via continual learning.
>
> When needed, retrieved long-term information is reintroduced into the robot’s short-term memory and processed alongside current observations to inform action prediction. This design is inspired by classical models of human memory \[1\]. **Importantly, a high-capacity short-term memory remains necessary for rapidly integrating recent sensory-motor experience and retrieved information for decision-making.**
>
> We have clarified this conceptual distinction and its role in our framework in the revised manuscript.

---

> > ### Author Response · Authors · 2025-11-23
> >
> > > Q1: The paper categorizes short-term memory into four types (spatial, prospective, object-associative, object-set) based on cognitive science literature. Are these four categories exhaustive, or are they a representative subset of a broader taxonomy? Could you clarify whether REMEMBENCH aims to cover all major categories of short-term memory relevant to robotics, or if additional categories might be added in future work? What criteria from cognitive science guided the selection of these specific four categories?
> >
> > The four categories do not constitute an exhaustive taxonomy of short-term memory. Instead, they represent a grounded subset, chosen by taking direct inspiration from established experimental paradigms in cognitive science that study human short-term memory (details in Sec. 4).
> >
> > The benchmark is **extensible**. Future versions can add new axes or their combinations, inspired by other human memory tests. Examples include:
> >
> > 1. **Temporal-order memory**: remembering the order in which events or instructions occur.
> > 2. **Relational memory**: comparing recent, similar experiences and leveraging their relationships to make better predictions
> >
> > The selection of the current four categories was guided by three principles:
> >
> > 1. **Direct grounding in well-studied human memory constructs** (spatial working memory, prospective memory, associative memory, and multi-object working memory), rather than ad-hoc task design.
> > 2. **Encouraging memory architectures that generalize across diverse demands**: using distinct categories (spatial, prospective, object-associative, object-set) pushes models toward general mechanisms rather than overfitting to a single memory type.
> > 3. **Embodied realizability with clear success criteria** in a realistic kitchen environment (navigation \+ manipulation with natural distractors).
> >
> > > Q2: Several tasks might potentially be solved without explicit memory through richer state representations. For example, in "Wash and Return to Container" (Fig.4), an observation function that detects wetness could disambiguate "just washed" vs. "needs washing" states, replacing the need to remember the washing action. Similarly, in "Cook Meat," a visual cue indicating stove temperature or cooking duration might substitute for remembering when the stove was turned on. This raises a fundamental question: what is the principled distinction between tasks that inherently require memory vs. tasks solvable through better perception/state engineering or other alternative techniques?
> >
> > Thank you for raising this important question. We agree that, in some cases, richer state representations could reduce reliance on explicit memory. However, even if perfect indicators were hand-engineered for one aspect of the state, the robot would still require memory for other critical dependencies. For example, in Wash and Return to Container, a wetness indicator could distinguish “just washed” from “about to wash,” but the robot must still remember which container the object originally came from and where that container was placed. Similarly, in Wash and Return to Same Location, the robot must recall the approximate location on the table where the object was first picked up, which is no longer directly observable after the scene has changed.
> >
> > More broadly, replacing memory with state engineering would require an ever-growing set of task-specific indicators, ranging from fine-grained properties such as container identity, container pose, wetness, and time since last interaction, to higher-level variables such as task progress and outcomes of prior attempts. Designing and maintaining such features across diverse objects, layouts, and tasks for general-purpose robots quickly becomes intractable. This mirrors trends in visual recognition, where hand-crafted features such as SIFT and HOG \[2,3\] were ultimately surpassed by end-to-end learned representations \[4\].
> >
> > We have added this discussion to the revised manuscript to clarify our motivation for focusing on learned memory mechanisms rather than increasingly complex hand-engineered state representations.
> >
> > ### References
> > \[1\] Atkinson & Shiffrin, *Human memory: A proposed system and its control processes*, 1968\.
> > \[2\] Lowe, “Distinctive Image Features from Scale-Invariant Keypoints,” IJCV 2004\.
> > \[3\] Dalal and Triggs, “Histograms of Oriented Gradients for Human Detection,” CVPR 2005\.
> > \[4\] Krizhevsky et al., “ImageNet Classification with Deep Convolutional Neural Networks,” NeurIPS 2012\.
> >
> > We have incorporated the changes in the manuscript and are happy to further clarify any remaining questions or concerns. We sincerely appreciate the reviewer’s time and consideration and would be grateful for any reassessment in light of the additional clarifications and results.

---

> ### Comment · Reviewer_1AkL · 2025-11-25
>
> Thank you for your response. Here are my follow-up comments:
>
> **Regarding Q1**, I recommend designing experiments that explicitly test **temporal memory capabilities**. Here is a concrete example:
>
> **Proposed experiment:**
> - Place three buttons of different colors (e.g., red, green, blue) in front of the robot
> - Present colored balls to the robot's camera in a specific sequence (e.g., red → green → blue)
> - Task: The robot should then press the buttons in the same order as the balls were presented
>
> This would test whether your pipeline can retain and recall temporal sequences. Could you please: (1) Implement this or a similar temporal memory task? (2) Compare your method against relevant baselines? (3) Include the ablation from **W1** to isolate the contribution of specific components?
>
> **Regarding Q2**, You mentioned that my suggested "state engineering" approach might be impractical. Could you please provide: (1) Experimental comparisons between explicit state engineering methods and your approach? (2) Quantitative or qualitative results demonstrating the advantages of your method over manually engineered state representations?

---

> > ### Author Response · Authors · 2025-12-03
> >
> > **Comparison with state-engineered baseline.** We thank the reviewer for their thoughtful suggestion. Designing a state-engineered baseline in ReMemBench is cumbersome because the benchmark includes diverse tasks that require capturing various aspects in state from past interactions, e.g., time tracking for prospective memory tasks, past object-object relations for object-associative tasks, spatial locations for spatial tasks, tracking set counts for object-set tasks, and semantics to distinguish among distractors. To closely follow the spirit of state engineering, we adapt the recent SAM2Act++ framework \[1\] to explicitly construct a compact spatial state representation: the center-pixel coordinates of previously observed objects. We additionally include the absolute timestep as a temporal feature. We concatenate these with the current observation and feed the resulting sequence into a vanilla transformer, which we refer to as the SAM2Act-like baseline. Even under these conditions, PRISM achieves an average $+11$ points higher success. SAM2Act-like, with temporal embedding and vanilla transformer backbone, performs especially poorly on tasks with distractors (e.g., **0%** on one Spatial Memory task), reflecting the limitations of spatial-only state features without semantic understanding. Likewise, its low performance on Prospective Memory tasks (**8%**) suggests that absolute timestamps fail to capture the relative temporal dependencies between the important events (e.g., turn on vs. turn off stove timesteps) required for success.
> >
> > These results illustrate a core challenge in ReMemBench: **hand-engineering state representations that generalize across heterogeneous memory requirements is fundamentally limited**, whereas PRISM learns to represent and retrieve the necessary information automatically.
> >
> > Evaluation on eight tasks of ReMemBench
> >
> > | Method | Spatial | Prospective | Obj-Associative | Obj-Set | Avg. |
> > | :---- | :---- | :---- | :---- | :---- | :---- |
> > | SAM2Act-like | 0.14 | 0.08 | **0.24** | 0.30 | 0.19 |
> > | PRISM | **0.37** | **0.35** | 0.15 | **0.33** | **0.30** |
> >
> > **Adding new tasks to ReMemBench.** We appreciate the reviewer’s suggestion. ReMemBench includes a substantially more diverse set of memory-centric manipulation tasks than prior benchmarks, spanning multiple categories, with an expert-teleoperated dataset (See Table 1). For instance, a recent benchmark on short-term memory with imitation learning, MemoryBench \[1\], includes only three tasks (vs. eight in ReMemBench) spanning two categories (vs. four in ReMemBench). While we agree that additional categories would further enrich the benchmark, expanding the suite requires new task designs, expert demonstrations, training, and evaluations. We view ReMemBench as an open and extensible platform and plan to incorporate new tasks in future releases, informed by valuable feedback such as this.
> >
> > \[1\] F. Haoquan, et al. "Sam2act: Integrating visual foundation model with a memory architecture for robotic manipulation." *ICML*, 2025\.

---

### Official Review · Reviewer_oRxT · 2025-10-31

**Soundness:** 2
**Presentation:** 3
**Contribution:** 2
**Rating:** 4
**Confidence:** 3

**Summary:**

This paper proposes PRISM, a transformer-style visuomotor policy that scales short-term memory for long-horizon manipulation via two components: (i) feature-aware gating that selectively filters history tokens based on their relevance, and (ii) a hierarchical memory that locally compresses tokens and performs global attention over compact summaries, reducing cost for long contexts. The authors introduce REMEMBENCH, eight kitchen-style manipulation tasks spanning four short-term memory categories—spatial, prospective, object-associative, and object-set—constructed so that success depends on within-episode recall rather than single-frame observability. PRISM is trained with imitation learning (IL) under partial observability from closed-loop observations (RGB + proprioception). Empirically, PRISM outperforms LSTM, Mamba, and a PTP-trained baseline on REMEMBENCH, shows some gains on unrelated atomic tasks, and demonstrates a modest but meaningful improvement on a real-robot task. Efficiency plots indicate substantially lower VRAM usage and latency at long context lengths versus non-hierarchical models.

**Strengths:**

* **Problem framing.** Treats IL under partial observability explicitly, with tasks that isolate different short-term memory demands.
* **Design.** Pragmatic combination of feature-aware gating and hierarchical summaries that targets the compute/latency pain of long contexts.
* **Empirics.** Consistent improvements on memory-dependent tasks; efficiency gains at inference; some real-robot evidence.

**Weaknesses:**

* **Missing transformer IL baselines.** The paper compares PRISM only to LSTM, Mamba, and a PTP-trained baseline, but omits transformer-based IL policies tailored to partial observability (e.g., windowed causal Transformers, Transformer-XL/GTrXL, linear-attention Transformers). Without these, it is unclear whether PRISM outperforms standard transformer sequence models for IL-POMDPs.
* **Memory architecture vs action space.** The authors note that many prior embodied works used discrete actions (true, e.g., Scene Memory Transformer). However, the key issue here is memory under partial observability, not the action parameterization. Transformer memory mechanisms—windowed, segment-recurrent (Transformer-XL/GTrXL), global/episodic (scene/episodic memory), and linear-attention—are action-agnostic and directly apply to continuous-action IL by swapping only the output head (Gaussian or diffusion). Omitting these transformer memory baselines—even with continuous heads—leaves the necessity of PRISM’s gate+hierarchy untested.
* **Prior-work coverage.** Related work should explicitly cover transformer memory for embodied agents (e.g., scene/episodic memory transformers) and segment-recurrent transformers for partial observability. In vision-language navigation (VLN), Episodic Transformer encodes the entire episode history (observations/actions + language), showing that long-horizon transformer memory is crucial for compositional tasks. Although the present method does not use language and outputs continuous actions, these architectures are action- and modality-agnostic and should be acknowledged and positioned.

**Questions:**

1. **Transformer baselines (continuous actions).** Please add:
   **(a)** Windowed Transformer BC; **(b)** Transformer-XL/GTrXL BC (segment length S, memory M); **(c)** a linear-attention Transformer BC.
   Use continuous outputs (Gaussian or diffusion), identical inputs/loss to PRISM, and match parameters, history length, and training budgets.
2. **Scene-memory variant.** On a small subset (e.g., one task per memory category), include an SMT-style global-memory baseline (global attention over stored observation embeddings) to bound accuracy/compute trade-offs relative to PRISM’s hierarchy.

---

> ### Author Response · Authors · 2025-11-23
>
> We thank the reviewer for their thoughtful feedback and for highlighting our problem framing (IL under partial observability with targeted memory tasks), pragmatic design (feature-aware gating with hierarchical architecture), and empirical gains (improved performance and efficiency).
>
> In response to the request for stronger transformer-based and memory-aware baselines, we have added the following comparisons on ReMemBench. All methods use identical continuous action outputs, identical inputs and loss functions, matched parameter counts, and the same training pipeline and budget (384 H100 GPU-hours per experiment).
>
> | Method | Spatial | Prospective | Associative | Object-Set | Average |
> | :---- | :---- | :---- | :---- | :---- | :---- |
> | TrXL | 0.06 | 0.28 | 0.12 | 0.12 | 0.15 |
> | GTrXL | 0.20 | 0.30 | **0.30** | 0.24 | 0.26 |
> | Linear-Attention | 0.10 | 0.40 | 0.20 | 0.18 | 0.22 |
> | Transformer BC | 0.12 | 0.44 | 0.20 | 0.20 | 0.24 |
> | PRISM w/o Gating | 0.08 | 0.35 | **0.30** | 0.30 | 0.26 |
> | SMT-style | **0.50** | 0.4 | 0.26 | 0.28 | 0.36 |
> | PRISM | 0.40 | **0.80** | 0.25 | **0.35** | **0.45** |
>
> > **W1: Missing transformer IL baselines.** The paper compares PRISM only to LSTM, Mamba, and a PTP-trained baseline, but omits transformer-based IL policies tailored to partial observability (e.g., windowed causal Transformers, Transformer-XL/GTrXL, linear-attention Transformers). Without these, it is unclear whether PRISM outperforms standard transformer sequence models for IL-POMDPs.
> **Q1: Transformer baselines (continuous actions).** Please add: **(a)** Windowed Transformer BC; **(b)** Transformer-XL/GTrXL BC (segment length S, memory M); **(c)** a linear-attention Transformer BC. Use continuous outputs (Gaussian or diffusion), identical inputs/loss to PRISM, and match parameters, history length, and training budgets.
>
> We have added comparisons with (a) Windowed causal Transformer BC, (b) Transformer-XL (TrXL), (c) Gated Transformer-XL (GTrXL), and (d) Linear-Attention. Note that a naive Windowed Transformer requires a very large context window to solve these tasks. Without the hierarchical architecture, it is computationally expensive (Figure 6, main paper), so we report the best numbers achieved with the same computation budget as PRISM, i.e., 384 H100 GPU hours per experiment.
>
> All added baselines consistently underperform PRISM on ReMemBench. We attribute this primarily to well-known limitations of these architectures in long-horizon settings: recurrent models such as TrXL ($-30%$) and GTrXL ($-19%$) suffer from credit assignment over long sequences due to backpropagation through time \[1,2\], while Linear-Attention exhibits substantial forgetting ($-23%$) \[3\]. Additionally, windowed causal Transformers without a hierarchical design become computationally prohibitive, leading to fewer effective gradient updates within the same budget. Removing the gating mechanism from PRISM results in a $-19\\%$ absolute drop. Together, these results isolate PRISM’s key contributions: improved retrieval via feature-wise gating and efficient computation via a hierarchical architecture.
>
> > **Q2: Scene-memory variant.** On a small subset (e.g., one task per memory category), include an SMT-style global-memory baseline (global attention over stored observation embeddings) to bound accuracy/compute trade-offs relative to PRISM’s hierarchy.
>
> To address this, we added an SMT-style global memory baseline that forms a single trajectory embedding via learned pooling over stored observations. While computationally efficient, it often over-compresses temporally critical information, losing task-relevant details and hurting performance ($-9\\%$). This performance drop suggests that a single pooled memory is insufficient, and that PRISM’s hierarchical memory with stacked layers is better suited for long-horizon visuomotor control.

---

> > ### Author Response · Authors · 2025-11-23
> >
> > > **W2: Prior-work coverage.** Related work should explicitly cover transformer memory for embodied agents (e.g., scene/episodic memory transformers) and segment-recurrent transformers for partial observability [...] should be acknowledged and positioned.
> > **W3: Memory architecture vs action space.** [...] Omitting these transformer memory baselines—even with continuous heads—leaves the necessity of PRISM’s gate+hierarchy untested.
> >
> > We thank the reviewer for highlighting important directions in transformer-based memory for embodied agents. In the revised version, we expand the related work section to explicitly cover segment-level recurrent transformers and scene memory transformers used in embodied AI and vision-and-language navigation. Architectures such as SMT compress full episode histories of observations and actions into a single representation via a global attention layer. In our experiments, we find that such compressed, global representations are insufficient for fine-grained, long-horizon visuomotor control (see below), motivating PRISM’s hierarchical and feature-wise memory design, which retrieves and selectively filters relevant information over extended horizons. Note that all methods use identical continuous action outputs, identical inputs and loss functions, the same parameter count, and the same training pipeline and budget.
> >
> > ### References
> > \[1\] Bengio, Yoshua, et al. "Credit assignment through time: Alternatives to backpropagation." *Advances in Neural Information Processing Systems*, 1993\.
> > \[2\] Tiezzi, Matteo, et al. "State-space modeling in long sequence processing: A survey on recurrence in the transformer era." *Neural Networks*, 2025\.
> > \[3\] He, Mutian, et al. "Alleviating Forgetfulness of Linear Attention by Hybrid Sparse Attention and Contextualized Learnable Token Eviction." *arXiv:2510.20787*, 2025\.
> >
> > We have incorporated the changes in the manuscript and are happy to further clarify any remaining questions or concerns. We sincerely appreciate the reviewer’s time and consideration, and would be grateful for any reassessment in light of the additional clarifications and results.

---

### Official Review · Reviewer_3P7K · 2025-11-01

**Soundness:** 2
**Presentation:** 3
**Contribution:** 2
**Rating:** 2
**Confidence:** 5

**Summary:**

A new transformer architecture, PRISM, and a robotics benchmark for memory-dependent tasks, ReMemBench, are proposed.

**Strengths:**

**Strengths:**
1. Memory-augmented architectures for robotics are an important and rapidly developing direction.
2. The paper is well-written and easy to read.
3. A new architecture and benchmark for memory-dependent robotic tasks are presented.
4. Experiments were conducted on a real robot.

**Weaknesses:**

**Weaknesses:**
1. There is no comparison of ReMemBench with the Mikasa-Robo benchmark [1], which is specifically designed to test memory mechanisms in robotics.
2. Only simple baselines, not designed for memory-dependent tasks, are used for comparison. Comparisons with specialized baselines, such as RATE [2] and GTrXL [3], are necessary.
3. The main experiments are conducted only on the benchmark proposed in the paper. Comparisons on existing benchmarks, such as Mikasa-Robo [1] and MemoryBench [4], are needed.
4. To ensure that the memory mechanism does not impair the model on simple tasks, evaluation on existing benchmarks, such as RLBench [5], LIBERO [6], and SimplerEnv [7], is critically important. Current results on RoboCasa are insufficient, especially since they are presented without the context of other approaches’ performance.
5. Detailed information on data collection and model training, which are necessary for reproducibility, is missing.

**Minor issues:**
1. Figure 5 overlaps with the text – line 422
2. Broken link to CrossMAE – line 676

**References:**
1. Cherepanov, Egor, et al. "Memory, Benchmark & Robots: A Benchmark for Solving Complex Tasks with Reinforcement Learning." arXiv preprint arXiv:2502.10550 (2025).
2. Cherepanov, Egor, et al. "Recurrent action transformer with memory." arXiv preprint arXiv:2306.09459 (2023).
3. Parisotto, Emilio, et al. "Stabilizing transformers for reinforcement learning." International conference on machine learning. PMLR, 2020.
4. Fang, Haoquan, et al. "Sam2act: Integrating visual foundation model with a memory architecture for robotic manipulation." arXiv preprint arXiv:2501.18564 (2025).
5. James, Stephen, et al. "Rlbench: The robot learning benchmark & learning environment." IEEE Robotics and Automation Letters 5.2 (2020): 3019-3026.
6. Liu, Bo, et al. "Libero: Benchmarking knowledge transfer for lifelong robot learning." Advances in Neural Information Processing Systems 36 (2023): 44776-44791.
7. Li, Xuanlin, et al. "Evaluating real-world robot manipulation policies in simulation." arXiv preprint arXiv:2405.05941 (2024).

**Questions:**

1. How does the task-type categorization in ReMemBench correspond to the memory types in Mikasa-Robo [1]?
2. How does the approach perform compared to strong baselines such as RATE [2] and GTrXL [3]?
3. How and in what quantity was the data collected for training?
4. Do the training details in Section B of the appendix apply only to PRISM, or to all baselines?

---

> ### Author Response · Authors · 2025-11-23
>
> We thank the reviewer for their thoughtful feedback and for highlighting the importance of memory-augmented architectures for robotics, as well as the clarity of the paper, the new architecture and benchmark, and the real-robot experiments. Below, we address each of the reviewer’s concerns and provide the suggested analyses and experimental results.
>
> >  Q1: How does the task-type categorization in ReMemBench correspond to the memory types in Mikasa-Robo \[1\]?
>
> We thank the reviewer for highlighting this recent and relevant benchmark. ReMemBench differs from Mikasa-Robo in several important ways:
>
> **Learning paradigm:** ReMemBench, with its human-teleoperated demonstrations, targets imitation learning under partial observability, whereas Mikasa-Robo primarily focuses on reinforcement learning.
>
> **Task scope and horizon:** ReMemBench spans both navigation and manipulation, involves multi-stage tasks, and operates over substantially longer horizons. Success typically requires recalling information beyond the initial frames, rather than simply “remembering the beginning.”
>
> **Memory taxonomy:** Mikasa-Robo organizes memory into four types: object, spatial, sequential, and capacity. ReMemBench overlaps with the object and spatial dimensions but additionally emphasizes object-associative and prospective memory, which are not explicitly captured in Mikasa-Robo. These categories target the recall of relationships between entities and the remembering of future-relevant information, thereby increasing the diversity of memory mechanisms under evaluation.
>
> **Environment complexity:** ReMemBench uses realistic kitchen environments with natural visual distractors, whereas Mikasa-Robo uses more controlled, task-centric simulated settings.
>
> These design choices lead to different performance trends across architectures, as reflected in our results. We have added an explicit discussion of Mikasa-Robo in the revised version.
>
> > W2 Only simple baselines, not designed for memory-dependent tasks, are used for comparison. Comparisons with specialized baselines, such as RATE \[2\] and GTrXL \[3\], are necessary.
> > Q2 How does the approach perform compared to strong baselines such as RATE \[2\] and GTrXL \[3\]?
>
> In response to the reviewer’s suggestion, we expanded our evaluation on ReMemBench to include several additional memory-focused architectures, specifically a linear-attention transformer (Linear-Attention), a segment-level recurrent transformer (TrXL), Gated Transformer-XL (GTrXL), and the Scene Memory Transformer (SMT). As in the original submission, we also include comparisons to Past Token Prediction (PTP) and non-transformer baselines such as LSTM and Mamba. The table below summarizes all baselines, with the first four rows covering recurrent architectures and the next four transformer-based ones. PRISM consistently outperforms all models, **including a $+19\\%$ improvement over GTrXL**, highlighting the benefit of combining learned gating with attention over both recurrent and standard transformer-based sequence models.
>
> Each experiment on ReMemBench requires roughly **384 H100 GPU-hours**, so a fully exhaustive study over additional recurrent architectures, such as **RATE,** would demand substantially more compute. We view a systematic comparison with RATE as valuable future work, but beyond the practical scope of this rebuttal.
>
> | Method | Spatial | Prospective | Associative | Object-Set | Average |
> | :---- | :---- | :---- | :---- | :---- | :---- |
> | LSTM | 0.14 | 0.04 | 0.12 | 0.04 | 0.09 |
> | Mamba | 0.04 | 0.34 | 0.07 | 0.08 | 0.13 |
> | TrXL | 0.06 | 0.28 | 0.12 | 0.12 | 0.15 |
> | GTrXL | 0.20 | 0.30 | 0.30 | 0.24 | 0.26 |
> | Linear-Attention | 0.10 | 0.40 | 0.20 | 0.18 | 0.22 |
> | PTP | 0.14 | 0.34 | 0.06 | 0.06 | 0.15 |
> | Standard Transformer | 0.12 | 0.44 | 0.2 | 0.2 | 0.24 |
> | Scene Memory Transformer | **0.50** | 0.4 | 0.26 | 0.28 | 0.36 |
> | PRISM | 0.40 | **0.80** | **0.25** | **0.35** | **0.45** |

---

> > ### Author Response · Authors · 2025-11-23
> >
> > > W1: There is no comparison of ReMemBench with the Mikasa-Robo benchmark \[1\], which is specifically designed to test memory mechanisms in robotics.
> > W3 The main experiments are conducted only on the benchmark proposed in the paper. Comparisons on existing benchmarks, such as Mikasa-Robo \[1\] and MemoryBench \[4\], are needed.
> >
> > In response to the reviewer’s request, we also evaluate PRISM on Mikasa-Robo. The results can be found on the website at [this link](https://remembench-prism.github.io/static/images/mikasa_robo_grid.png) or in Appendix Figure 7 of the updated manuscript. While PRISM improves performance on some tasks, such as InterceptSlow-v0 and RememColor5-v0, its overall performance is comparable to LSTM-based baselines on this benchmark. This contrasts with ReMemBench, where transformer-based models significantly outperform recurrent architectures (Table 2). We hypothesize this difference arises because Mikasa-Robo tasks (i) have substantially shorter horizons, approximately 180 timesteps compared to 2,800 timesteps in ReMemBench, (ii) contain fewer natural visual distractors, and (iii) concentrate most task-relevant information in the initial frames. As a result, Mikasa-Robo places less emphasis on sustained short-term memory over long horizons, which is the primary regime targeted by ReMemBench.
> >
> > MemoryBench \[4\] primarily targets a narrow form of spatial memory under controlled settings. In the original benchmark paper, SAM2Act+ achieves 84–100% success across its three tasks: Reopen Drawer ($84\\%$), Put Block Back ($100\\%$), and Rearrange Block ($99\\%$). These near-ceiling results leave limited headroom for differentiating methods or evaluating more general or long-horizon memory mechanisms. In contrast, ReMemBench comprises 8 tasks designed to probe broader short-term memory demands, including spatial, prospective, object-set, and object-associative memory, which are better aligned with the requirements of general-purpose robots.
> >
> > > W4: To ensure that the memory mechanism does not impair the model on simple tasks, evaluation on existing benchmarks, such as RLBench \[5\], LIBERO \[6\], and SimplerEnv \[7\], is critically important. Current results on RoboCasa are insufficient, especially when presented without the context of other approaches’ performance.
> >
> > To verify that our memory mechanism does not degrade performance on simpler tasks, we expanded our evaluation on RoboCasa by adding strong baselines (BC-Transformer, Diffusion Policy, and GR00T-N1-2B) and further report results across all LIBERO benchmarks. PRISM outperforms its no-memory variant by $+11\\%$ on RoboCasa and $+12\\%$ on LIBERO. It also surpasses strong pretrained baselines, including an absolute gain of $ \+11\\%$ over the large-scale VLA model GR00T-N1-2B finetuned on RoboCasa and $+16\\%$ over OpenVLA finetuned on LIBERO.
> >
> > These consistent gains show that short-term memory not only avoids harming performance on seemingly memory-less tasks but also actively improves it by providing additional task-relevant context and reducing action multimodality. While we cannot evaluate on other additional benchmarks due to computational constraints, the results on ReMemBench, RoboCasa, and LIBERO demonstrate that PRISM is broadly beneficial across both explicitly memory-dependent tasks and tasks that appear largely memory-less.
> >
> > | Method | RoboCasa (Success Rate) |
> > | :---- | :---- |
> > | BC-Transformer | 0.26 |
> > | Diffusion Policy | 0.26 |
> > | GR00T-N1-2B (pretrained \+ finetuned) | 0.32 |
> > | PRISM (no memory) | 0.32 |
> > | PRISM (n=256) | **0.43** |
> >
> > | Method | LIBERO 90 | LIBERO 10 | LIBERO Object | LIBERO Spatial | LIBERO Goal |
> > | :---- | :---- | :---- | :---- | :---- | :---- |
> > | OpenVLA | 0.62 | 0.54 | 0.88 | 0.85 | 0.79 |
> > | Diffusion Policy | \- | 0.72 | **0.93** | 0.78 | 0.68 |
> > | PRISM (no memory) | 0.77 | 0.75 | 0.89 | 0.85 | 0.61 |
> > | PRISM (n=256) | **0.85** | **0.81** | **0.93** | **0.92** | **0.95** |
> >
> > > Detailed information on data collection and model training, which are necessary for reproducibility, is missing.
> >
> > We appreciate the reviewer’s emphasis on reproducibility. We commit to releasing the complete codebase. Model training details are provided in Appendix Section B, and training hyperparameters are in Table 3\.
> >
> > > How and in what quantity was the data collected for training?
> >
> > Training data were collected via human teleoperation: 50 trajectories per task, each spanning roughly 400-2800 timesteps. The details are included in the revised version of the manuscript (Section B).
> >
> > > Do the training details in Section B of the appendix apply only to PRISM, or to all baselines?
> >
> > They apply to **all** models. For fairness, we use the same training pipeline, optimizer, and hyperparameters for PRISM and baselines.
> >
> > We have incorporated the changes in the manuscript and are happy to further clarify any concerns. We appreciate the reviewer’s effort and would be grateful for any reassessment in light of the clarifications and results.

---

### Official Review · Reviewer_G3H6 · 2025-11-03

**Soundness:** 3
**Presentation:** 3
**Contribution:** 2
**Rating:** 6
**Confidence:** 4

**Summary:**

This manuscript brings a new benchmark `ReMemBench` and also solves the long context limitation of the transformer based policies by introducing an attention mechanism.

**Strengths:**

New Benchmark is provided.

Manuscript is easy to read and all the steps are sound and intuitive.

Real-robot deployment.

State of the art results.

**Weaknesses:**

Methodological novelty is minimal. For example, using attention mechanism to remove distractions in test time in imitation learning for visuomotor policies has been demonstrated before. For example: `“Pay attention!-
robustifying a deep visuomotor policy through task-focused visual
attention,” CVPR, 2019`.

**Questions:**

1- How the model can decide if everything from a timestamp is relevant and needs to be tokenized or part of it. And if this is configurable or automated (learned) by the model. What is the impact of it? Meaning what if we have stricter attention mechanism that is very conservative in selecting information, vs having a more free flow of information into the memory? How does it impact the performance on the benchmarks?

2- I am wondering if the authors tried applying noise to the attention mechanism during training to make it more resilient during test time.

3- [Out of curiosity, authors can skip this question] Can the proposed method be extended to novel objects during test time? How much drop we expect if the task requires manipulating a novel object, not seen in training.

4- Line 249: I am wondering if the design of g(x) like number of parameters or other designs rather than simple MLP can impact the final results.

---

> ### Author Response · Authors · 2025-11-23
>
> We thank the reviewer for recognizing the significance of our contributions, including the introduction of a new benchmark, the clarity and soundness of our presentation, the real-robot validation, and the strong empirical performance of our approach. Below, we address each of the reviewer’s concerns and provide all the suggested analyses and experimental results.
>
> > **Methodological novelty is minimal. For example, using attention mechanism to remove distractions in test time in imitation learning for visuomotor policies has been demonstrated before. For example: “Pay attention\!- robustifying a deep visuomotor policy through task-focused visual attention,” CVPR, 2019\.**
>
> Thank you for highlighting “Pay Attention\!” (CVPR 2019), which we have added to our related work section. That approach uses an LSTM to integrate temporal information; however, as our experiments show, LSTMs and other recurrent architectures struggle with credit assignment over long horizons, and in our benchmark, LSTM and Mamba baselines underperform transformer variants by 32–36 percentage points (Table 2). We therefore focus on transformer-based visuomotor policies and investigate how to equip them with effective short-term memory for long-horizon tasks.
>
> **Methodological novelty.** Rather than simply extending a transformer’s context window, which introduces spurious correlations and prohibitive compute/memory costs, we introduce (i) a post-retrieval, feature-wise gating module that suppresses irrelevant information and (ii) a hierarchical memory architecture that factorizes expensive attention into cheaper stages, enabling successful imitation on two-minute-long tasks, an order of magnitude longer than prior work \[5–6\]. To evaluate such mechanisms under partial observability, we introduce ReMemBench, a cognitively grounded benchmark spanning four distinct types of short-term memory, on which our method significantly outperforms state-of-the-art baselines.
>
> > 1- How the model can decide if everything from a timestamp is relevant and needs to be tokenized or part of it. And if this is configurable or automated (learned) by the model.
>
> PRISM stores all sensory inputs at 5 Hz, similar to standard transformer-based policies, without requiring manual timestep selection. Instead, relevance is determined during attention-based retrieval. The network first attends to candidate tokens via standard query-key matching, followed by a learned, post-retrieval feature-wise gating module that our experiments show is essential for filtering irrelevant information over long horizons (Figure 5, middle). The entire process is fully automated and trained end-to-end with the action-prediction loss, without any hand-crafted token-selection rules.
>
> > What is the impact of it? Meaning what if we have stricter attention mechanism that is very conservative in selecting information, vs having a more free flow of information into the memory? How does it impact the performance on the benchmarks?
>
> We explored stricter information flow by sharpening the attention distribution (i.e., scaling the attention logits so that fewer tokens carry most of the probability mass). More conservative information flow (larger scaling factors) led to a slight performance degradation ($-2%$ to $-10%$), while overly permissive attention ($\\text{factor} \= 0.125$) resulted in a large performance drop ($-42%$). These results indicate that controlling information flow is critical, a role explicitly handled by PRISM’s learned post-retrieval gating module rather than relying solely on attention.
>
> | $scale\\\_factor$\*$head\\\_dim^{1/2}$ | 0.125 | 0.25 | 0.5 | 1.0 (PRISM) | 2.0 | 4.0 | 8.0 |
> | :---- | :---- | :---- | :---- | :---- | :---- | :---- | :---- |
> | Retrieve Fruit | 0.22 | 0.54 | 0.54 | 0.64 | 0.64 | 0.54 | 0.62 |
>
> > 2- I am wondering if the authors tried applying noise to the attention mechanism during training to make it more resilient during test time.
>
> To evaluate robustness, we injected noise into the attention mechanism by randomly dropping tokens with probability $p=0.1$ \[7\]. This improves PRISM’s performance by  $+4%$. We also applied the same token-drop strategy to PRISM without gating. While both variants benefit from noisy attention, the full PRISM architecture continues to outperform the no-gating variant by $+10%$ (with attention drop) and $+19%$ (without attention drop). These results indicate that noise alone is helpful but insufficient, and that the learned gating module is crucial for suppressing spurious correlations.
>
> | Method | Spatial | Prospective | Associative | Object-Set | Average |
> | :---- | :---- | :---- | :---- | :---- | :---- |
> | PRISM w/o Gating | 0.08 | 0.35 | **0.30** | 0.30 | 0.26 |
> | PRISM w/o Gating (Attn Drop) | 0.44 | **0.80** | 0.10 | 0.20 | 0.39 |
> | PRISM | 0.40 | **0.80** | 0.25 | 0.35 | 0.45 |
> | PRISM (Attn Drop) | **0.60** | 0.64 | **0.30** | **0.40** | **0.49** |

---

> ### Author Response · Authors · 2025-11-23
>
> > 3- \[Out of curiosity, authors can skip this question\] Can the proposed method be extended to novel objects during test time? How much drop we expect if the task requires manipulating a novel object, not seen in training.
>
> We thank the reviewer for this suggestion and evaluated generalization to novel objects on the Retrieve Fruit task by training on object categories (e.g., banana and strawberry) and testing on unseen categories (e.g., apple and pear). We observe a $12\\%$ performance drop, from $64\\%$ to $52\\%$, indicating that object-level generalization remains a challenge. While this is not the paper's primary focus, we plan to include a “novel object” variant in the final release of ReMemBench. We view closing this performance gap, for example, via pretrained vision-language-action models or broader multi-task training on RoboCasa, as promising future work and believe this addition will be valuable to the community for studying generalization in long-horizon, memory-dependent tasks.
>
> > 4- Line 249: I am wondering if the design of g(x), like number of parameters or other designs rather than simple MLP can impact the final results.
>
> This is an important concern. The inevitable changes induced by g(x), such as adding parameters or non-linearities, could confound our claim that improvements stem from **filtering irrelevant information**.
>
> To understand the effect of the feature-wise gating mechanism from other confounding factors, we compare **blind gating** (86M parameters, sigmoid non-linearity) with **feature-wise gating** (90M parameters, sigmoid non-linearity) in Figure 5 (middle). Blind gating, instead of producing a per-feature gate, learns a single-value gate per layer and therefore has the same non-linearity and a similar number of parameters as PRISM (less than $5\\%$ extra). However, the performance gap between the two is stark: feature-wise gating yields a $**\+17\\%$** improvement, suggesting that the gains primarily arise from feature-wise information selection rather than other factors.
>
> | Method | Spatial | Prospective | Associative | Object-Set | Average |
> | :---- | :---- | :---- | :---- | :---- | :---- |
> | Blind Gating | 0.35 | 0.30 | 0.25 | 0.20 | 0.28 |
> | Feature Gating (PRISM) | **0.40** | **0.80** | **0.25** | **0.35** | **0.45** |
>
> ### References
> \[1\] Y. Bengio, et al, "Learning long-term dependencies with gradient descent is difficult," *IEEE Transactions on Neural Networks*, 1994.
> \[2\] C. Cheng, et al. "Diffusion policy: Visuomotor policy learning via action diffusion." *IJRR*, 2025\.
> \[3\] B. Johan, et al. "Gr00t n1: An open foundation model for generalist humanoid robots," *arXiv:2503.14734*, 2025\.
> \[4\] B. Jose, et al. "A careful examination of large behavior models for multitask dexterous manipulation." *arXiv:2507.05331*, 2025\.
> \[5\] F. Haoquan, et al. "Sam2act: Integrating visual foundation model with a memory architecture for robotic manipulation." *ICML*, 2025\.
> \[6\] T. Marcel, et al. "Learning Long-Context Diffusion Policies via Past-Token Prediction." *CoRL*, 2025\.
> \[7\] H. Le, et al. "Token dropping for efficient bert pretraining." *arXiv:2203.13240*, 2022\.
>
>
>
> We have incorporated the changes in the manuscript and are happy to further clarify any remaining questions or concerns. We sincerely appreciate the reviewer’s time and consideration, and would be grateful for any reassessment in light of the additional clarifications and results.

---

### Comment · Area_Chair_dgSs · 2025-11-24

Dear Reviewers,

The authors have responded to your reviews. For those who have not yet done so, please read the authors' comments and respond to them.

Best, Your AC

---

### Author Response · Authors · 2025-12-03

We thank the AC for their time. In response to the reviewer’s feedback, we have substantially expanded our experiments and analyses. Below, we summarize the key aspects of our work.

Our **goal** is to equip visuomotor policies with *short-term memory* for diverse, long-horizon robotic tasks.

**Proposed Method.** We propose **PRISM**, a transformer-based visuomotor policy trained with imitation learning to predict low-level actions directly from sensory inputs. PRISM can retrieve and leverage information from up to the past two minutes, **over 10× longer than prior work** \[1,2\], enabled by two key technical innovations:

1. **Feature-wise gating to suppress noise from retrieved information.** A post-attention gating mechanism selectively filters retrieved features, allowing the policy to *suppress irrelevant information* and avoid spurious correlations.
2. **Hierarchical architecture to reduce the computational and memory cost.** We decompose the expensive full-attention computation into two cheaper hierarchical computations: a local memory module (with compression factor $m$) and a global memory module, enabling us to scale short-term memory over a longer horizon.

**Proposed Benchmark.** Existing imitation-learning benchmarks for visuomotor control (e.g., RoboCasa, LIBERO, RLBench, Simpler) do not explicitly test partial observability and the short-term memory needed to handle it. To address this gap, we release **ReMemBench**, a long-horizon manipulation benchmark with a human-teleoperated dataset for imitation learning, grounded in four types of tests applied in cognitive science (spatial, prospective, object-associative, and object-set) (See Sec. 4 for benchmark details and Table 1 for comparison with other benchmarks).

**Strong Empirical Evidence.**

1. **PRISM achieves SOTA on ReMemBench** ($45\\%$ absolute success) \[Table 2\]
   1. recurrent baselines (\#*G3H6*): LSTM ($36$-point drop), Mamba ($32$-point drop)
   2. segment-level recurrent baselines (\#*3P7K*, \#*oRxT*): TrXL ($30$-point drop), Gated-TrXL ($19$-point drop)
   3. faster attention variants (\#*3P7K*, \#*oRxT*): linear-attention ($23$-point drop)
   4. other prior architectures proposed in visuomotor policies with imitation learning (\#*oRxT*, \#*1AkL*): PTP ($30$-point drop), Scene Memory Transformer ($9$-point drop), SAM2Act-like ($11$-point drop)
   5. For a detailed discussion on the results, refer to Section 5.1.
2. **PRISM also shows strong transfer to real-world tasks** with a gain of $15$-point drop ($30\\%$ vs. $15\\%$ absolute success) over PTP \[2\] in tasks, where no memory baseline leads to 0% success.
3. **Gains observed even on seemingly “memoryless” benchmarks: LIBERO and RoboCasa** (Figure 5 Left, Tables 7 and 8 in the Appendix) (\#*3P7K*)
   1. On LIBERO and RoboCasa, PRISM outperforms its no-memory variant by $11$-point and $14$-point gain, respectively.
   2. **Outperforms strong finetuned VLA baselines, without any pretraining.**
      1. $11$-point gain over GR00T-N1-3b (finetuned on RoboCasa
      2. $16$-point gain over OpenVLA (finetuned on LIBERO)
4. Ablations validating design choices.
   1. **The gating mechanism is essential.** Our proposed feature-wise gating outperforms all alternatives, including (Figure 5, Middle):
      1. per-layer gating value ($11$-point drop)
      2. no gating ($13$-point drop)
      3. attention sinks ($29$-point drop)
   2. **Hierarchical architecture yields efficiency gains** (\#*1AkL*). Compared to full attention or hybrid attention variants, PRISM’s hierarchical design reduces:
      1. peak GPU memory by $80\\%$
      2. runtime by $40\\%$
   3. **The compression factor controls resource utilization** (\#*1AkL*). Increasing the compression factor ($m=2 \\rightarrow 200$)
      1. peak GPU memory by $56\\%$
      2. runtime by $13\\%$
   4. **Attention sharpness analysis shows that attention alone is brittle** (\#*G3H6*)**.** We observe that,
      1. Adding irrelevant details through diffused attention leads to a drop in performance ($10$ to $42$-point drop)
      2. Overly sharp attention reduces useful retrieval capacity ($2$ to $10$-point drop)
5. **Results on MIKASA-Robo (**\#*3P7K***).** On Mikasa-Robo, PRISM outperforms LSTM on some tasks, though overall trends are closer to LSTM baselines (Appendix C.4). This trend sharply contrasts with ReMemBench and highlights how task diversity, horizon length, and the learning setup can shape trends across different memory architectures, underscoring the need for complementary benchmarks like ReMemBench. See Appendix Section C.4 for more details.

\[1\] F. Haoquan, et al. "Sam2act: Integrating visual foundation model with a memory architecture for robotic manipulation." *ICML*, 2025\.
\[2\] T. Marcel, et al. "Learning Long-Context Diffusion Policies via Past-Token Prediction." *CoRL*, 2025\.

---

### Meta-Review · Area_Chair_v2cN · 2026-01-07

**Summary:**

One reviewer acknowledged the empirical strength of the results and real-robot validation, the majority expressed significant reservations. Critics pointed to limited technical novelty. Several reviewers questioned whether the gains stem from architectural choices or simply increased model capacity and extensive tuning. Overall, the paper did not meet the acceptance criteria.

**Reviewer Concerns:**

The authors provided thorough responses to all raised weakness and questions.

**Reviewer Scores:**

Reviewer G3H6 (initial 6) would likely maintain their score, as their moderate concerns were addressed but no indication of a score increase was given.
Reviewer 3P7K (initial 2) would likely keep their original score, as their fundamental concerns about novelty and benchmark design were only partially alleviated by the rebuttal.
Reviewer oRxT (initial 4) would probably retain their score, since while technical clarifications were provided, they did not express intent to upgrade their assessment.
Reviewer 1AkL (initial 2) would almost certainly maintain their low rating.

---

### Decision · Program_Chairs · 2026-01-26

Reject